# Prolyl hydroxylase-dependent proteolysis enables the orthogonal hypoxia responses in plants

Vinay Shukla[1,7], Sergio Iacopino[1,2,7], Laura Dalle Carbonare[1], Alessia Del Chiaro[1], Yuming He [1], Mauricio Nicolàs Tronca [1], Thomas P. Keeley[3,4,5], Antonis Papachristodoulou [6], Beatrice Giuntoli [2] ✉ & Francesco Licausi [1] ✉

Vascular plants and metazoans use selective proteolysis to control responses to hypoxia, although through distinct biochemical mechanisms. The reason for this divergence is puzzling, since the molecular components necessary for both strategies are conserved. To explore the alternative scenario where plants and animals respond to hypoxia through the same mechanisms, we engineer a three-components system aimed to target proteins for degradation in an oxygen dependent manner *in Arabidopsis thaliana*. When used to control transcription, the synthetic system partially restores hypoxia responsiveness in oxygen-insensitive mutants. Additionally, we demonstrate its potential to regulate growth under flood-induced hypoxia. Our work highlights the use of synthetic biology to reprogramme signalling pathways, providing insights into the evolution of oxygen sensing and offering tools for crop improvement under stress conditions.

Aerobic organisms require molecular oxygen ($O_2$) for sufficient ATP synthesis to support growth and development. When $O_2$ provision to cells from the surrounding environment is limited (hypoxia), cells adjust their structure, physiology and metabolism to avoid or resist the stress. Such adaptations require transcriptional reprogramming triggered by dedicated hypoxia-sensing mechanisms. Remarkably, vascular plants and metazoans share a similar sensing strategy based on oxygen-dependent degradation of constitutively expressed transcription factors (TFs) and their nuclear accumulation upon exposure to hypoxia[1]. However, this is achieved by different biochemical solutions in the two kingdoms.

In vascular plants, a major role in transcriptional reprogramming is played by the Ethylene Response Factor VIIs (ERFVIIs), TFs controlled via the Plant Cysteine Oxidase (PCO)-branch of the N-degron pathway[2–4]. This is a proteasomal degradation pathway that dictates protein stability depending on the exposed N-terminal amino acid[5].

Methionine amino peptidases prepare the ERFVIIs to expose an N-terminal cysteinyl residue for N-terminal sulfinylation via PCOs[6]. This modification promotes N-terminal arginylation, which is in turn recognised by the ubiquitin E3 ligase PROTEOLYSIS6 (PRT6), with the assistance of BIG, for polyubiquitination and the consequent degradation through the proteasome[7–9] (Supplementary Fig. 1a).

Although mammalian cells have an oxygen-sensing system based on N-cys oxidation, they regulate transcriptional reprogramming with a different mechanism, relying on the Hypoxia-Inducible Factors (HIFs)[10,11]. They are dimers of α and β subunits, the former controlled via oxygen-dependent degradation[12,13]. This regulation is mediated by the hydroxylation of internal prolyl residues, catalysed by $O_2$ and 2-oxoglutarate (2-OG)-dependent Prolyl-hydroxylases (PHDs)[14,15]. The hydroxylated HIFα subunit is recognised by a ubiquitin E3 ligase complex through the von Hippel-Lindau factor (VHL)[16,17]. As in the case of the ERFVII, polyubiquitinated HIFα is degraded by the proteasome.

[1]Department of Biology, University of Oxford, South Parks Road, Oxford, United Kingdom. [2]Department of Biology, University of Pisa, Via Luca Ghini 13, Pisa, Italy. [3]Department of Physiology, Anatomy and Genetics, University of Oxford, Oxford, United Kingdom. [4]Target Discovery Institute, Nuffield Department of Medicine, University of Oxford, Oxford, United Kingdom. [5]Ludwig Institute for Cancer Research, Nuffield Department of Medicine, University of Oxford, Oxford, United Kingdom. [6]Department of Engineering Science, University of Oxford, Parks Road, Oxford, United Kingdom. [7]These authors contributed equally: Vinay Shukla, Sergio Iacopino. ✉e-mail: beatrice.giuntoli@unipi.it; francesco.licausi@biology.ox.ac.uk

When $O_2$ levels drop, instead, HIFα is protected from proteolysis and thereby accumulates in the nucleus. Here, heterodimerization with its cognate β-subunit reconstitutes a functional transcription complex able to induce expression of hypoxic genes (Supplementary Fig. 1b).

The evident similarity between $O_2$-sensing strategies from vascular plants and metazoans has led to the speculation that selective proteolysis represents the best solution to accommodate organised multicellularity[18]. However, the last common ancestor of plants and animals likely expressed both PHD and N-Cys dioxygenase[18,19]. This raises the question as to whether PHD adoption better suits $O_2$ sensing in heterotrophs with an active system for gas circulation, whereas N-Cys dioxygenases optimally accommodate photosynthetic organisms. The recruitment of oxygenases with distinct kinetic parameters as $O_2$ sensors could reflect substantial differences in the responses produced in the two kingdoms under hypoxia, in the $O_2$ levels at which these need to be activated and the existence of peculiar mechanisms of interference or crosstalk with metabolites and secondary messengers[20–22].

Hypotheses regarding the origin and differentiation of molecular mechanisms are traditionally addressed by comparisons across extant species. We reasoned that a synthetic biology approach would instead tackle more directly the question about the convergence/divergence of $O_2$-sensing mechanisms in animals and plants[23]. Therefore, we set out to reconstruct in plant cells a hypoxia-responsive mechanism based on proline hydroxylation and subsequent proteolysis. Previously, we exploited the $O_2$-dependent interaction of HIF and VHL to engineer a molecular circuit useful to inhibit gene expression in response to hypoxic inputs[24]. Here, we re-engineered $O_2$-dependent proteolysis to control the stability of reporters and signal transducers. Besides addressing the fundamental question regarding the evolution and diversification of $O_2$ sensing in multicellular eukaryotes, we considered that such an effort would serve as a proof of concept for the engineering of proteostatic signalling in plant cells[25]. Moreover, by doing so, we thought to establish an orthogonal switch to control the response to environmental conditions that limit $O_2$ availability for plants, such as submergence.

Flooding is one of the main causes for agricultural losses worldwide[26]. Since gas diffusion is severely reduced in water, submerged plant organs experience limited $O_2$ availability. Tampering with the endogenous $O_2$ sensing system is likely to compromise plant fitness overall. This is because the N-cys degron pathway controls the response to different environmental stresses including cold, high salinity, pathogen attack and dehydration[27–29] and also participates to developmental processes[30–33]. We therefore applied the synthetic biology framework to generate an orthogonal switch for transcriptional regulation in response to hypoxia in plant cells. We used this molecular device to explore the alternative evolutionary scenario where $O_2$ sensing in plants depends on PHDs and tested its ability to drive developmental responses to submergence-induced hypoxia.

## Results

### Engineering PHD-dependent proteolysis in plants

We set out to reconstruct an orthogonal $O_2$-dependent switch for plant cells by inducing targeted proteolysis based on the mammalian HIF-1α/VHL system (Supplementary Fig. 1B). Using the model species *Arabidopsis thaliana*, we have shown previously that the C-terminal Oxygen-dependent degron of HIF-1α ($HIF_{ODD}$) and VHL β-domain (aa 63-157) interact in an $O_2$-dependent manner, exclusively when a human PHD3 enzyme is expressed[24]. Given the 1.7–1.4 billion years evolutionary distance between plants and animals[34], we considered it unlikely that the mammalian VHL protein retained the ability to associate with functional E3 ligase complexes in Arabidopsis cells. An initial comparison of in silico reconstructed Cullin-Ring E3 ligase (CRLs) complexes from animals and plants suggested similar subunit arrangements between the complex responsible for HIF-α

hydroxylation in human cells and those dedicated to auxin signalling in Arabidopsis (Fig. 1A). However, while VHL contacts the CRL complex via the interaction between its α-domain with ElonginB (ELOB) and ELOC[35] (Fig. 1A), the auxin sensor Transport Inhibitor Response1 (TIR1) does so by binding to the Arabidopsis homologue of the mammalian S-phase kinase associated protein 1 (Skp1), Ask1[36] (Fig. 1A). The α-domain of VHL resembles TIR1 F-box, with a similarly oriented three-helical structure. Nonetheless, VHL's inability to contact Ask1 in Arabidopsis cells is suggested by the distinct amino acid residues involved in VHL/ELOC and F-box/Ask1 interaction, coupled with other minor structural differences (Supplementary Fig. 2A, B), and by the absence of a Cullin2 homologue in plants[37]. Therefore, we decided to replace the original α-domain of VHL with an endogenous F-box domain to enable the incorporation of the chimeric VHL in a CRL E3 ligase complex. Under aerobic conditions, and with the expression of a human PHD3 enzyme[24], the chimeric F-box protein would be expected to bind a hydroxylated $HIF_{ODD}$ peptide, thereby promoting the ubiquitination and degradation of the fusion protein in which it is contained. Hypoxia, instead, should prevent recognition and thus stabilise the $HIF_{ODD}$-tagged protein (Fig. 1B).

We first attempted to generate the desired chimeric F-box protein by fusing the F-box domain of TIR1 to the VHL β-domain. To monitor HIF degradation, we produced a chimeric transcription factor consisting of the $HIF_{ODD}$ peptide, the minimal GAL4 DNA binding domain and the small but potent activation domain of the Arabidopsis transcription factor Related to APETALA2.12 (RAP2.12)[38] (Fig. 1C). This construct was expected to bind the synthetic 4xUAS promoter[24] and to promote the transcription of a Firefly Luciferase (FLuc) reporter gene (Fig. 1C). When expressed in Arabidopsis protoplasts, $HIF_{ODD}$-GAL4-AD indeed induced FLuc reporter activity, but this was not significantly affected by the presence of VHL-F-box and PHD3 (Supplementary Fig. 3A, B). This result suggested the inability of the TIR1-based chimeric VHL to contact the $HIF_{ODD}$ peptide, independent of its hydroxylation state. Therefore, we tested a second F-box domain from the SLEEPY1 (SLY1) protein, one of the smallest functional F-box proteins in the Arabidopsis proteome[39], which participates in gibberellin (GA) signalling[40]. Two gibberellin-insensitive SLY1 variants, SLY1-2 and SLY1-10, have been characterised as early protein truncations that abolish gibberellin responsiveness while maintaining intact the F-box domain[40] (Supplementary Fig. 4A). Since no crystal structure for SLY1 is available, we superimposed the predicted model of its F-box domain on TIR1 to verify the conserved identity and position of residues necessary for the interaction with Ask1 (Supplementary Fig. 4B). We fused the wild type and truncated SLY1 variants at the N-terminal end of VHL β-domain and tested them in protoplast, as described before for TIR1. This time, combination of SLY1-VHL with PHD3 abolished FLuc induction by $HIF_{ODD}$-GAL4-AD, indicating the efficacy of the three chimeric F-box versions at ubiquitinating the hydroxylated transcription factor (Fig. 1D). We selected the smallest variant, SLY1-2-VHL, for further experiments, after validating its nuclear localisation (Supplementary Fig. 4C), under the assumption that it would interfere the least with GA signalling. Hypoxia alleviated the repression imposed by SLY1-2-VHL on transcription factor activity (Fig. 1E), indicating that $HIF_{ODD}$-GAL4-AD degradation was inhibited when insufficient $O_2$ availability limited PHD3 activity. This hypothesis was corroborated by immunodetection of GAL4-AD in a western blot (Fig. 1F).

### PHD-dependent conditional proteolysis is transferable to different reporter proteins

The newly established $O_2$-dependent proteolytic mechanism could be used to control the stability of other proteins to generate novel types of genetically encoded reporters of hypoxia in plants. To implement this application, we generated single transcriptional units coding for two luciferases (hereafter Luc1 and Luc2, based on their position in the construct) separated by a ubiquitin unit so that, during mRNA

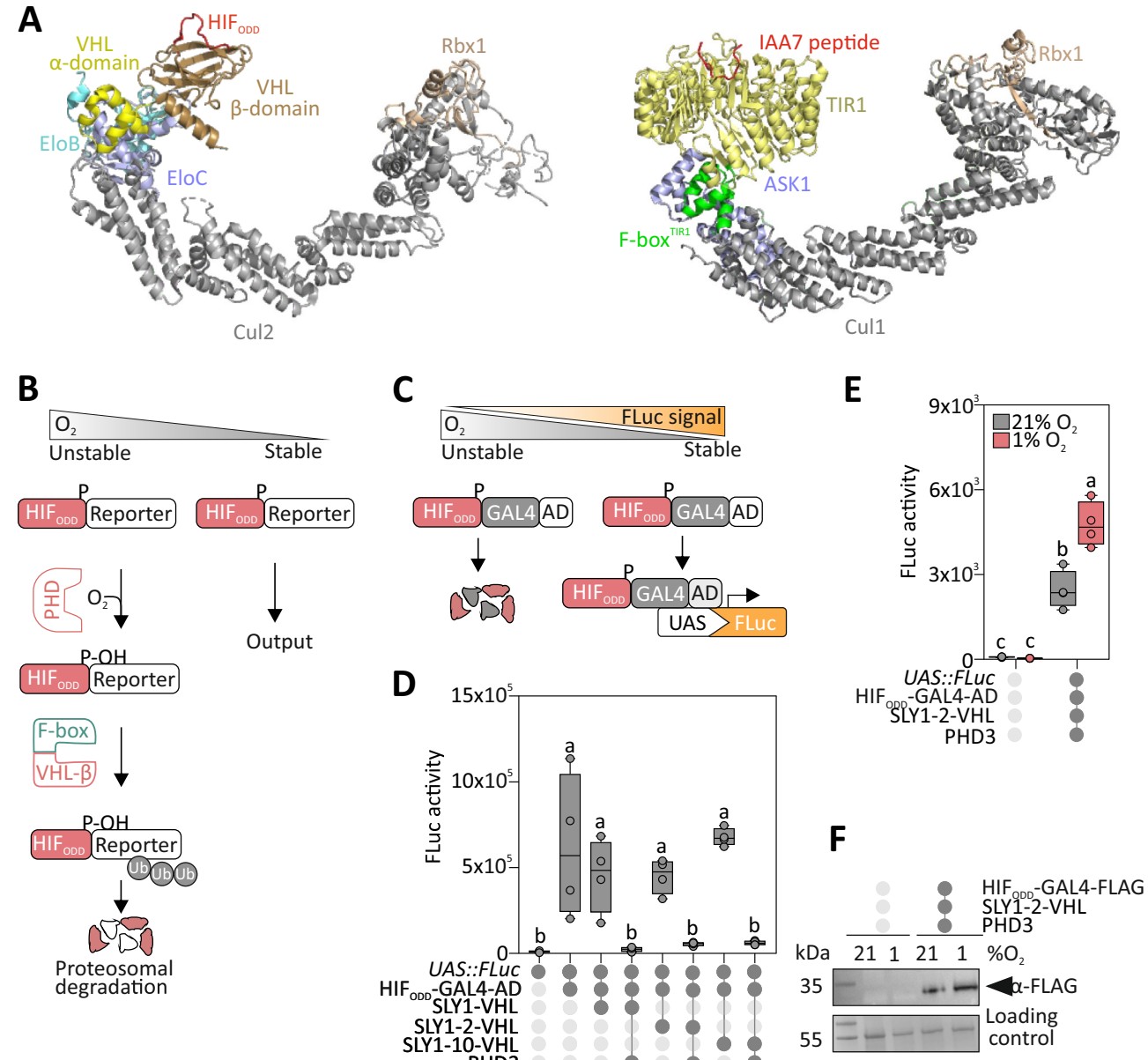

**Fig. 1 | Implementation of PHD- and VHL-dependent $O_2$ sensing in plant cells.** **A** Comparison of the predicted human VBC-CR (left) and plants SCF[Tir1] (right) complexes. The VBC-CR complex comprises the scaffold protein Cul2 (grey), the E3 ubiquitin ligase Rbx1 (light brown), the adaptor proteins EloB (aquamarine) and EloC (blue) and the substrate recognition protein VHL (α-domain in yellow, β-domain in bronze). The plant SCF complex consists of the Cul1 scaffold protein (grey), Rbx1 (light brown), the adaptor Ask1 (blue) and the substrate recognition F-box protein TIR1 (LRR region in yellow, F-box domain in green). The peptides shown for the animal and plant substrates are HIF$_{ODD}$ and IAA7, respectively. **B** The $O_2$-dependent proteolytic mechanism engineered for plant cells. A chimeric E3 ligase, which comprises the VHL beta domain and a plant F-box domain, recognises the HIF$_{ODD}$ domain when this is hydroxylated by PHD in the presence of $O_2$. Reporter proteins linked to the HIF$_{ODD}$ are degraded in normoxia but stable in hypoxia. **C** Reporter strategy deployed to test the synthetic $O_2$ sensing system. HIF$_{ODD}$ was fused to the DNA-binding domain of GAL4 and the activation domain (AD) of RAP2.12. In the presence of $O_2$, this chimeric transcription factor is degraded by the proteasome. In hypoxic conditions, instead, it is stabilised and binds the *4xUAS* promoter to activate transcription of a FLuc reporter. **D** FLuc output measured in Arabidopsis mesophyll protoplasts transformed with combinations of plasmids to express HIF$_{ODD}$-GAL4-AD, three variants of the SLY1-VHL chimera (Supplementary Fig. 4A) and PHD3. Letters indicate statistically significant differences ($p \leq 0.05$), as determined by one-way ANOVA followed by Tukey's post-hoc test ($n = 4$ independent biological replicates, each replicate consisting of protoplasts isolated from an independent batch of plants and measured as the average of technical luciferase readings). Data are shown as box plots, where the centre line indicates the median, box bounds indicate the 25th and 75th percentiles, whiskers extend to the minimum and maximum values, and all points represent individual biological replicates. **E** Effect of hypoxia (1% $O_2$, 6 h) on FLuc output in protoplasts expressing HIF$_{ODD}$-GAL4-AD, SLY1-2-VHL and PHD3. Letters indicate statistically significant differences ($p \leq 0.05$), as determined by two-way ANOVA followed by Tukey's post-hoc test ($n = 4$ independent biological replicates, each replicate consisting of protoplasts isolated from an independent batch of plants and measured as the average of technical luciferase readings). Data are shown as box plots, where the centre line indicates the median, box bounds indicate the 25th and 75th percentiles, whiskers extend to the minimum and maximum values, and all points represent individual biological replicates. **F** Immunodetection of HIF$_{ODD}$-GAL4-AD using a anti FLAG antibody in the same experimental setup as (**E**).

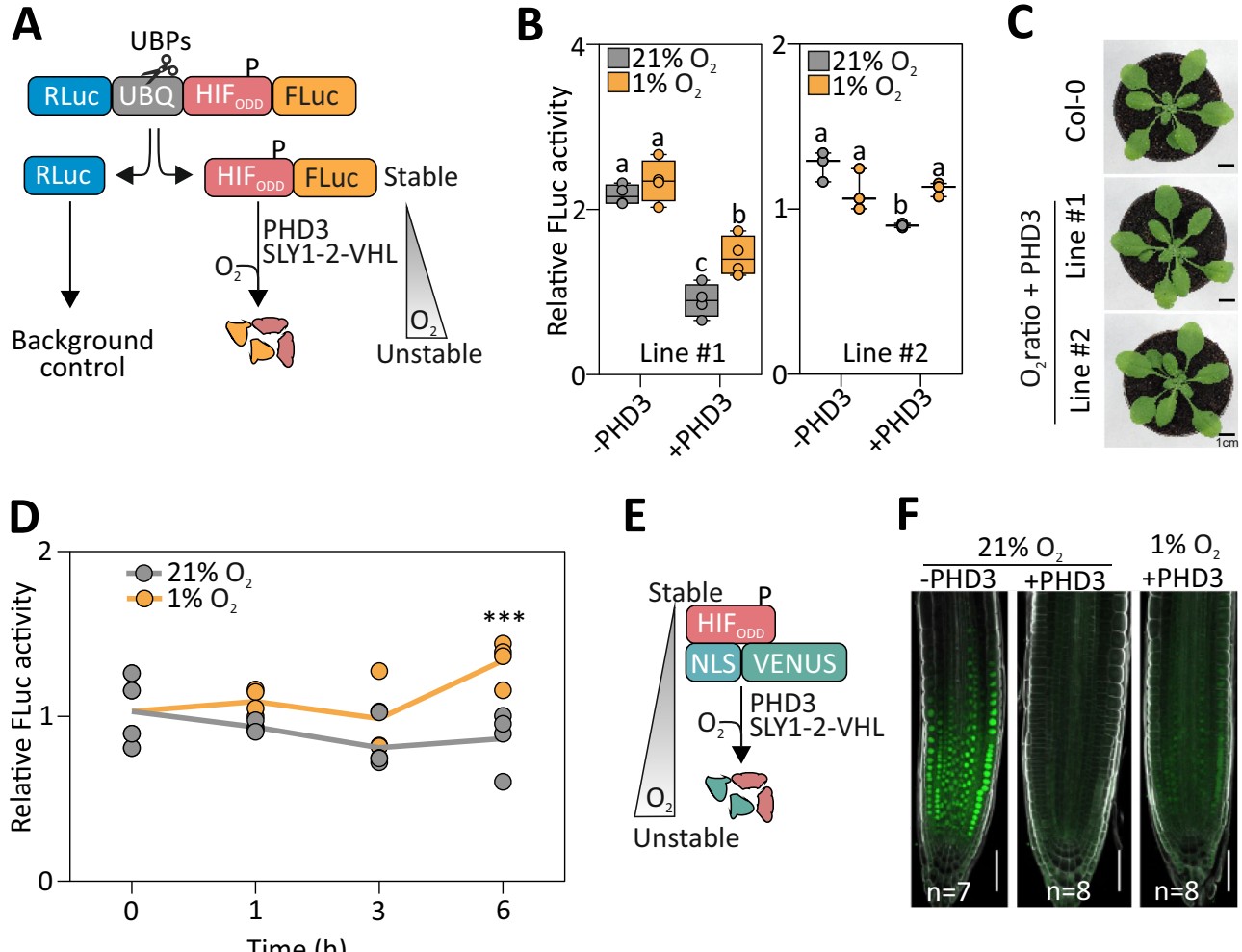

**Fig. 2 | Design and testing of a ratiometric PHD- and VHL-dependent O₂ reporter in plants. A** O₂ratio, a polyprotein consisting of Renilla Luciferase (RLuc), a UBQ10 unit and FLuc-HIF_ODD, was expressed in plant cells together with PHD3 and SLY1-2-VHL (not shown). Endogenous ubiquitin proteases (UBPs) co-translationally cleave the polypeptide at the C-terminus of UBQ10. The resulting Rluc serves as a normalisation reference while FLuc-HIF_ODD is used to monitor O₂ levels. **B** PHD- and O₂-dependent stability of O₂ratio in protoplasts that express the modules described in A. Letters indicate statistical differences ($p \leq 0.05$) determined by two-way ANOVA followed by Tukey's post-hoc test ($n = 4$ independent biological replicates, each replicate consisting of protoplasts isolated from an independent batch of plants and measured as the average of technical luciferase readings). Data are shown as box plots, where the centre line indicates the median, box bounds indicate the 25th and 75th percentiles, whiskers extend to the minimum and maximum values, and all points represent individual biological replicates. **C** Phenotype of 4-week-old O₂ratio and wild-type plants (scale – 1 cm). Representative image of 5 observations. **D** O₂ratio output dynamics in *35S:PHD3* O₂ratio seedlings during a hypoxia (1% O₂, yellow) or normoxia (21% O₂, grey) time course. The relative luminescence output was normalised to the average FLuc/RLuc value at time 0 h. Letters indicate statistical difference ($p \leq 0.05$) between treatments at each time point, as assessed by one-way ANOVA followed by Tukey's post-test ($n = 4$). **E** The fluorescent protein Venus, fused with a nuclear localisation signal (NLS) and HIF_ODD, serves as a reporter of O₂ levels. **F** Stability of the Venus fluorescence signal when regulated by PHD activity and atmospheric O₂ availability in root tips of 7-day old Arabidopsis seedlings subjected to 21% or 1% O₂ for 6 h (scale – 50 µm). Representative image of 7 observations.

translation, deubiquitinases (DUBs) would release two fragments, as described by Bachmair et al.[41]. The Luc1-UBQ fragment is suitable for signal normalisation whereas the stability of the HIF_ODD-fused Luc2 depends on O₂ and PHD3 (Supplementary Fig. 5A). We prepared several variants of this construct, distinguished by the combination of three features: the 5′ untranslated mRNA region (UTR), the paired luciferases and the number of HIF_ODD motifs associated with Luc2. We generated two independent transgenic lines of Arabidopsis stably expressing the SLY1-2-VHL effector along with each reporter variant. Mesophyll protoplasts isolated from each line were then transformed with a *35S:PHD3* construct for testing, or a *35S:GFP* control construct (Supplementary Fig. 5B). The desired reporter behaviour consisted of a decreased normoxic output upon PHD3 expression and high output recovery under hypoxia. As the best performing reporter of our set, we thereby identified a construct, driven by the *35S CaMV* promoter,

consisting of the *Renilla reniformis* luciferase (RLuc) as Luc1 and of FLuc as Luc2, equipped with a single C-terminal HIF_ODD domain (Fig. 2A, B). We named this O₂-dependent ratiometric construct O₂ratio. The other variants showed little responsiveness to either PHD3 or hypoxia, resulting in insufficient dynamic range (Supplementary Fig. 5C). To further expand the dynamic range of O₂ratio, we attempted a rational strategy of PHD mutagenesis aimed at lowering its enzymatic activity under low oxygen. Inspired by past studies on the human PHD2 enzyme[42–44], we tested three different single amino-acid substitutions in the transient protoplast system (Supplementary Fig. 6A, B). Nevertheless, the wild-type PHD3 showed the best performance and had a consistent behaviour across the two independent SLY1-2-VHL/O₂ratio lines used in the analysis (Supplementary Fig. 6C, D). We then proceeded to stable transformation the *35S:PHD3* construct in one of them and isolated two independent PHD3 lines

(Supplementary Fig. 6E, F). Neither of them showed developmental or growth differences in comparison with wild-type plants, indicating substantial orthogonality of the system (Fig. 2C). Looking at $O_2$ratio dynamics in *PHD3*-expressing plants under normoxia and hypoxia, we found that the earliest significant output increase occurred after 6 h of hypoxia (Fig. 2D).

Finally, we tested the portability of the regulation by transferring the $HIF_{ODD}$ degron to a nuclear VENUS reporter, expressed under the control of the p16 promoter (Fig. 2E)[45]. Using confocal microscopy, we could detect a strong fluorescent signal in the root tip of transgenic plants expressing this construct, which disappeared almost completely after expression of a *PHD3* transgene (Fig. 2F). This latter transgenic line recovered nuclear fluorescence when plants were incubated for 6 h at 1% $O_2$ (Fig. 2F). We concluded that the combination of three modules of human origin (*PHD3*, *SLY1-2-VHL* and $HIF_{ODD}$) is a broadly transferable system to control the stability of reporter proteins in plants depending on $O_2$-availability.

## N-cys oxidation and Pro hydroxylation are interchangeable to control hypoxia-dependent protein stability in plant and animal cells

The possibility of establishing $O_2$-dependent, targeted proteolysis allowed us to test an evolutionary scenario where hypoxia signalling did not diverge between vascular plants and metazoans. We did so by wiring the ERFVII transcription factor RAP2.12 to PHD-mediated degradation in Arabidopsis. To this end, we uncoupled RAP2.12 from the PCO-branch of the Arg/N-degron pathway by substituting its N-cys-degron (aa 2–13) with the $HIF_{ODD}$ peptide (Fig. 3A). This construct, controlled by the endogenous *RAP2.12* promoter, was expressed together with the *PHD3* and *SLY1-2-VHL* modules in a pentuple *erfVII* mutant[30]. We applied the same strategy used for $O_2$ratio to generate independent lines expressing $HIF_{ODD}$-ΔRAP2.12 and SLY1-2-VHL and super-transformed them with a *35S:PHD3* construct (Supplementary Fig. 7A). In this case, we used the promoter of the Arabidopsis gene *PCO4*[46] to control *PHD3*, to ensure similar expression levels of the $O_2$-sensing modules in the native plant mechanism and the synthetic system. $HIF_{ODD}$-ΔRAP2.12 stimulated the constitutive activation of the hypoxia marker genes *PCO1* and *SAD6;* co-expression of *PHD3* reversed this phenotype in normoxia but was counteracted by hypoxia (Supplementary Fig. 7B). Altogether, this indicated that the synthetic gene circuit is effective in controlling ERFVII activity. We set out to compare the performance of the endogenous and synthetic oxygen-sensing systems. Because the *Arabidopsis thaliana* genome codes for five ERFVIIs[47], we transformed the *erfVII* mutant with a full RAP2.12 coding sequence, driven by its endogenous promoter, to generate a control genotype with a single ERFVII. We tested the stability of the two RAP2.12 versions in a hypoxia time course by western blot, taking advantage of a fused double human influenza hemagglutinin (2xHA) tag. Wild-type RAP2.12 was barely detectable under normoxic conditions and transiently stabilised after 1–2 h hypoxia (Fig. 3B, Supplementary Fig. 8A). The $HIF_{ODD}$ version of RAP2.12 was more abundant than the native version under normoxia and further increased under hypoxia, although with variable dynamics across repetitions of the experiment (Fig. 3C, Supplementary Fig. 8B, C). Remarkably, its stability and also reproducibly increased when plants were kept in the dark for 4 h under aerobic conditions (Supplementary Fig. 8B, C). As additional validation, we verified that the instability of $HIF_{ODD}$-RAP2.12 is mediated by PHD3 through the proteasome. Loss of the PHD3 transgene enhanced $HIF_{ODD}$-RAP2.12 levels, which were further increased upon treatment of seedlings with the proteasome inhibitor bortezomib (BZ, 50 μM) (Fig. 3D).

These observations suggest a higher efficiency of the N-degron pathway in keeping ERFVII levels low under normoxia when compared with the newly engineered SLY1.2-VHL pathway. This could be explained as a refinement evolved over 450 million years, the time

estimated for the recruitment of ERFVIIs as N-Cys-degron substrates for $O_2$ sensing[48]. To test this hypothesis, we compared the efficiency of the two $O_2$-dependent proteolytic systems also in mammalian cells. We employed an immortalised human kidney proximal tubular cell line (HKC-8) previously edited with CRISPR to inactivate the endogenous HIF1α and HIF2α [49] (Supplementary Fig. 9A, B). Opposite to the wild type, these double knock-out (DKO) cells are unable to induce genes that carry a Hypoxia Responsive Element (HRE) in their promoter when exposed to hypoxia[49]. Mirroring our approach with the Arabidopsis RAP2.12, we aimed to obtain a version of HIF-1α that is no longer a substrate of PHDs but regulated in an $O_2$-dependent manner via the N-degron pathway (Fig. 3E). We produced three HIF-1α variants: a wild-type version, one carrying a double substitution on the two proline residues (HIF1α PPAA) that are hydroxylated by the PHD enzymes (P402A, P564A)[50] and a third version where HIF1α PPAA was equipped with the N-cys degron from the human Regulator of G-protein-signalling 4 (RGS4)[11] (Fig. 3E). We transfected the DKO cells with these constructs and evaluated HIF-1a protein expression under normoxia and hypoxia. The anti-HIF1α antibody used for immunoblotting generated a faint band in hypoxic untransformed cells, which likely corresponds to the truncated version produced by CRISPR (Fig. 3F, Supplementary Fig. 9C). As expected, wildtype HIF-1α was stabilised under hypoxia, whereas the PPAA version was already stable in normoxia (Fig. 3F, Supplementary Fig. 9C). Remarkably, addition of the $RGS4_{1-11}$ N-degron successfully restored HIF-1α PPAA $O_2$-dependent regulation (Fig. 3F, Supplementary Fig. 9C). We also used a dual luciferase reporter essay based on a synthetic 6xHRE promoter[51,52] to test how $O_2$-dependent control of HIF-1α protein abundance affect transcriptional regulation. Co-transfection of the luciferase reporter together with the empty vector control produced a minimal signal, in line with the lack of functional HIF-1α in DKO cells[49] (Supplementary Fig. 9D). In contrast, the presence of HIF-1α resulted in a strong increase of the signal, in an $O_2$-dependent manner for the wild-type, whereas the PPAA mutant activates the reporter to a comparable extent in normoxic and hypoxic conditions (Fig. 3G). Remarkably, the synthetic N-cys-degron version of HIF-1α had a comparable range of induction of the 6xHRE promoter to the wild-type, even though normoxic and hypoxic values were lower (Fig. 3G). Together with the protein abundance data (Fig. 3F), this observation confirmed that the N-degron pathway is more effective at repressing protein accumulation when compared with the VHL pathway, similar to what we observed in plant cells (Fig. 3B–D). Overall, we concluded that the N-degron pathway could be a valid alternative to PHD/VHL to adjust HIF-1α levels in response to the $O_2$ availability.

## PHD-dependent proteolysis can control transcriptional responses to hypoxia in plant cells

Having demonstrated that the PHD-dependent version of RAP2.12 is $O_2$-regulated in Arabidopsis, we moved on to compare the downstream transcription in response to hypoxia between the synthetic PHD/VHL-based and the native N-degron pathway-based $O_2$ sensing. The expression patterns of Hypoxia Responsive Genes (HRG) were first modelled by means of dynamic, ordinary differential equations (ODE). We incorporated in the equations both Michaelis-Menten Kinetics and Mass Action laws to describe the gene expression and biochemical reaction network processes (Supplementary Fig. S10A, B). The two models predicted early HRG activation upon exposure to 1% $O_2$ in both the native and synthetic context, with overall higher expression when ERFVIIs are controlled by the PHD/VHL degradation pathway (Fig. 4A). When HRG expression was simulated as a function of oxygen availability, both systems exhibited nonlinear oxygen-dependent behaviour. However, the response profiles differed markedly: HRG expression increased more gradually across oxygen levels in the PHD/VHL-based system, whereas the PCO-regulated system showed a steeper, more switch-like response (Fig. 4B).

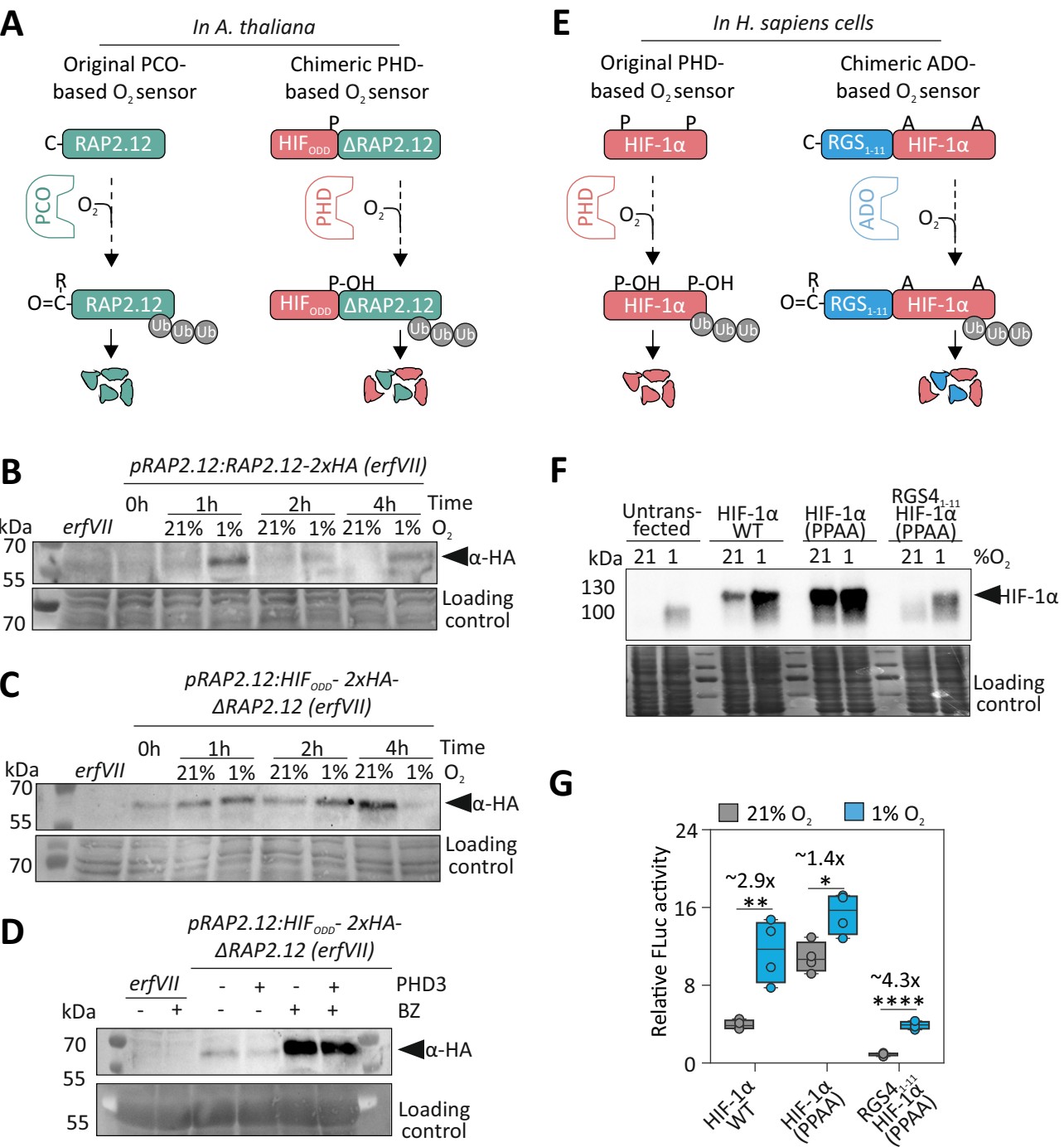

**Fig. 3 | Complementation of the hypoxia-insensitive *erfVII* mutant by synthetic PHD-based oxygen sensing. A** In *A. thaliana*, RAP2.12 stability is regulated in an $O_2$-dependent manner through the PCO/N-degron pathway (left). A chimeric RAP2.12 is instead controlled by the synthetic PHD/VHL regulatory module. To achieve this, the N-terminal 13 aa were removed from RAP2.12 (ΔRAP2.12) to bypass PCO regulation and substituted with HIF$_{ODD}$. Under normoxia, HIF$_{ODD}$-ΔRAP2.12 is expected to undergo PHD-dependent degradation. **B** Immunodetection of RAP2.12 and **C** of HIF$_{ODD}$-ΔRAP2.12 following a treatment with 1% $O_2$ for 4 h. Ponceau staining of total proteins is shown as the loading control. The *erfVII* background (1% $O_2$, 2 h) is shown as a negative control for both transgenic lines. The experiment was repeated independently 2 times with similar results. **D** Immunoblot detection of HIF$_{ODD}$-ΔRAP2.12 in 7-day-old seedlings expressing or not expressing human PHD3, following treatment with 50 µM bortezomib (BZ) or mock (0.02% DMSO). The experiment was repeated independently 2 times with similar results. **E** In mammalian cells, HIF-1α stability is regulated in an $O_2$-dependent manner through the PHD/VHL pathway (left). A chimeric HIF-1α is instead controlled by the ADO/N-

degron pathway (right). To achieve this, Pro402 and Pro564 were substituted with Ala to bypass PHD regulation, and the N-cys degron of RGS4 (aa1-11) was added. Under normoxia, RGS4$_{1-11}$- HIF-1α is expected to undergo ADO-dependent degradation. **F** Immunodetection of HIF-1α in DKO cells transiently transfected with plasmids coding for wild-type HIF-1α, HIF-1α(PPAA) and RGS4$_{1-11}$-HIF-1α(PPAA) and treated with 21% or 1% O2 for 16 h. Coomassie blue staining of total proteins is shown as the loading control. The experiment was repeated independently 2 times with similar results. **G** Fluc/Rluc signal at 21% $O_2$ and 1% $O_2$ (16 h) of transiently transfected DKO cells ($n = 4$ independent biological replicates, each replicate corresponding to an independently transfected well of HKC−8 DKO cells measured once for Firefly and Renilla luciferase). Statistical significance of differences was assessed by an unpaired two-tailed *t*-test (*: $p < 0.05$, **: $p < 0.01$, ***: $p < 0.001$, and ****: $p < 0.0001$). Data are shown as box plots, where the centre line indicates the median, box bounds indicate the 25th and 75th percentiles, whiskers extend to the minimum and maximum values, and all points represent individual biological replicates.

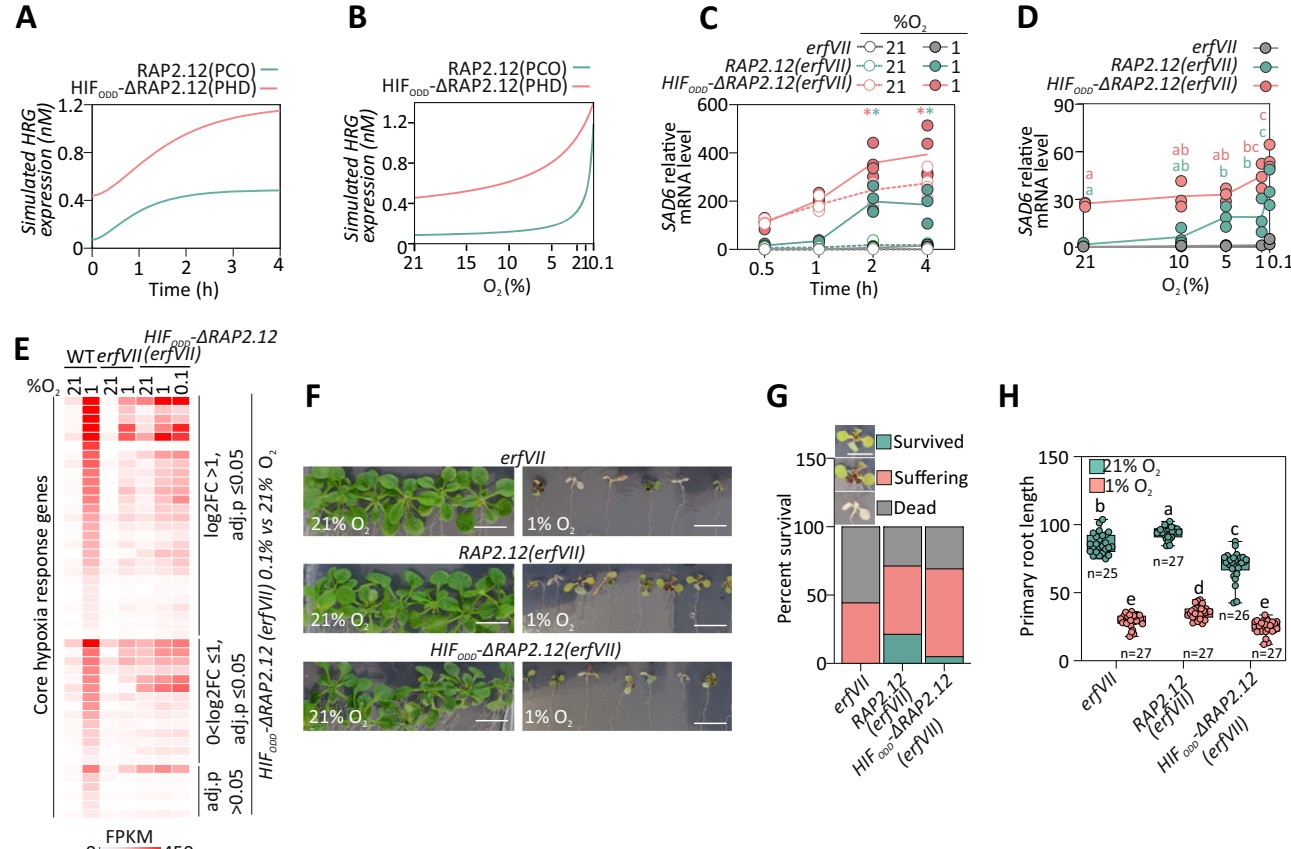

**Fig. 4 | Functional characterisation of native and synthetic O₂ sensing mechanisms. A, B** Predicted induction of a generic hypoxia-responsive gene (HRG) by RAP2.12 under the control of either the PCO oxygen-sensing mechanism or the PHD synthetic sensing pathway. **A** HRG expression over a simulated 4-h treatment at 1% (v/v) O₂. **B** HRG expression across a range of O₂ concentrations spanning from 21% to 0.1%. **C** Expression of the HRG *SAD6* in 7-day-old Arabidopsis *erfVII* seedlings and complementation lines expressing RAP2.12-HA (teal) or HIF_ODD-ΔRAP2.12-HA (pink) grown under normoxia and then subjected to 21 or 1% O₂ for 0.5 to 4 h. Statistical significance of hypoxic gene expression relative to 21% O₂ controls is denoted by asterisks (*t*-test, *p* ≤ 0.05). **D** Dynamics of *SAD6* marker gene expression in response to different O₂ levels in *erfVII* plants expressing RAP2.12-HA under control of the PCO or PHD pathways in an *erfVII* background. Letters indicate statistical significance of hypoxic treatments compared with 21% O₂ within each genotype, analysed independently by one-way ANOVA and followed by Tukey's post-hoc test (*p* ≤ 0.05). **E** Heatmap showing the effect of 2 h of hypoxia (at 1% or 0.1% O₂) in HIF_ODD-ΔRAP2.12-HA/*erfVII* plants on Fragments Per Kilobase of transcript per Million mapped reads (FPKM) of core hypoxia-response genes. Data for wild type and *erfVII* are taken from Dalle Carbonare et al.[48]. **F** Hypoxia tolerance assay of *erfVII* mutants and complemented lines expressing RAP2.12-HA or HIF_ODD-ΔRAP2.12-HA. Seven-day-old seedlings were grown on agar plates and exposed to 1% O₂ for seven days, followed by a seven-day recovery period. Images were taken at the end of the recovery period. **G** Plant tolerance after hypoxia (*n* ≥ 36). Inset images illustrate representative status for categorisation and quantification. **H** Primary root length of seedlings from (**D**) at the end of the 7-day hypoxia treatment. Letters indicate statistical significance, with two-way ANOVA and Tukey's post-test (*p* ≤ 0.05, n = number of individual seedlings per genotype and condition and is indicated below each box.). Data are shown as box plots, where the centre line indicates the median, box bounds indicate the 25th and 75th percentiles, whiskers extend to the minimum and maximum values, and all points represent individual biological replicates.

We therefore tested these predictions by comparing the induction of hypoxia marker genes in the same *erfVII* complemented plants described in Fig. 3B, C. At 1% O₂, the mutant *erfVII* background showed little residual capacity to elicit the expression of *PCO1* and *SAD6* (Fig. 4C, Supplementary Fig. 11A), as previously reported[48]. Both complementation strategies caused instead stronger expression of the reporter genes, with time-resolved induction dynamics similar to those predicted by our models (Fig. 4C, Supplementary Fig. 11A). We further compared the two O₂-sensing systems in an experimental setup where plants were exposed to a range of sub-atmospheric O₂ concentrations for 2 h. We observed that, while hypoxic gene expression was induced below 10% O₂ in the native N-degron context, in the PHD/VHL context, the response appeared only at 1% O₂ (Fig. 4D, Supplementary Fig. 11B). We expanded this analysis by looking at the entire set of 49 core HRGs[53] under hypoxia (1% O₂) and near-anoxia (0.1% O₂) in the HIFODD-RAP2.12 line using RNA-seq (Fig. 4E, Supplementary Data 1). Overall, the transgenic construct elevated HRG expression, which was

severely reduced in the *erfVII* background[48]. Despite the constitutively higher expression in normoxia, 55% of the core HRGs were significantly induced more than two-fold in the HIFODD-RAP2.12 line, while 30% showed a more modest, but still significant upregulation in hypoxia (Fig. 4E, Supplementary Data 2). We concluded that ERFVIIs retain the capacity to activate most HRGs under hypoxia when placed under the control of the PHD/VHL pathway, although with an attenuated dynamic range.

Finally, we compared the tolerance to the hypoxic stress of plants equipped with the two alternative O₂-sensing systems. We subjected 1-week old plants to 1% O₂ for 7 days, under photoperiodic conditions. Both O₂-sensing strategies were associated with decreased lethality as compared to the *erfVII* mutant, although the PCO-regulated RAP2.12 conferred higher fitness than the PHD-regulated version (Fig. 4F, G, Supplementary Fig. 12A, B). However, as previously reported[33], higher RAP2.12 abundance in the latter genotype impaired primary root growth, both under aerobic and hypoxic conditions (Fig. 4H).

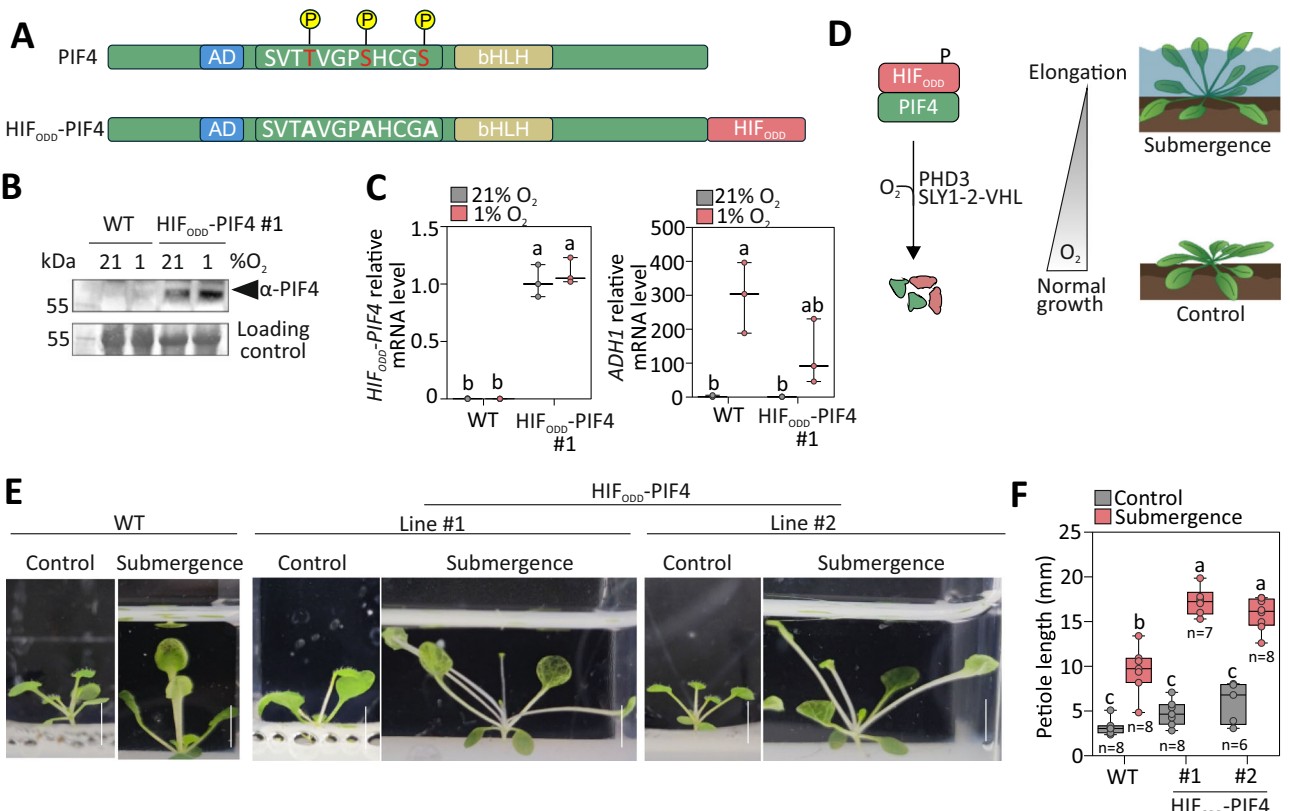

**Fig. 5 | Synthetic control of petiole elongation under flooding-induced hypoxia. A** Design of a PIF4-HIF$_{ODD}$ chimeric factor. Modifications applied to PIF4 to uncouple it from light, temperature and hormone regulation and rewire it to the mammalian-inspired synthetic $O_2$ sensing system devised in this study. **B** Immunodetection of *HIF$_{ODD}$-PIF4 in 7-day old seedlings* following a treatment with 21% or 1% $O_2$ for 6 h. Ponceau staining of total proteins is shown as the loading control. The Col-0 background is shown as a negative control. The experiment was repeated independently 2 times with similar results. **C** Relative mRNA levels of *HIF$_{ODD}$-PIF4 in 7-day old seedlings* following a treatment with 21% or 1% $O_2$ for 6 h. The Col-0 background is shown as a negative control. Data are mean ± s.d. of $n = 3$ independent biological replicates, each consisting of RNA extracted from an independent pool of seedlings. **D** Predicted regulation of *HIF$_{ODD}$-PIF4*. Under

flooding-induced hypoxia, the *HIF$_{ODD}$-PIF4* protein is stabilised, promoting petiole elongation. In normoxia and with active PHD3, *HIF$_{ODD}$-PIF4* is targeted for degradation via SLY1-2-VHL. **E** Phenotype of 2-week-old wild-type and PIF4-HIF$_{ODD}$ plants treated for 4 days with darkness (control) or dark submergence. **F** Quantification of petiole elongation from the plants in (**E**). Letters indicate statistical significance, with two-way ANOVA and Tukey's post-test ($p \leq 0.05$). Data are the mean individual petiole length of leaf number 4, per genotype and treatment, measured from independent plants. Data are shown as box plots, where the centre line indicates the median, box bounds indicate the 25th and 75th percentiles, whiskers extend to the minimum and maximum values, and all points represent individual biological replicates.

## Exploiting PHD-dependent proteolysis to engineer adaptive responses to flooding in plants

Once established that a PDH/VHL-based mechanism for hypoxia sensing can successfully drive transcriptional responses in plant cells, we explored whether the same could be connected to responses that increase plant fitness under $O_2$ limitation. Reduced $O_2$ availability commonly occurs because of soil waterlogging or whole plant submergence[26]. Flood-adapted species have evolved two contrasting strategies: those frequently experiencing long-term and shallow submergence enhance stem and/or petiole elongation to reach the water surface and thus maintain gas exchange ('escape'); those instead exposed to deep or short-lasting floods suppress organ elongation and metabolism ('quiescence')[54]. We attempted to promote either response through the synthetic PHD-dependent oxygen sensing. The Arabidopsis ecotype Col-0 exhibits modest organ elongation underwater[55]. To enhance its escape syndrome, we decided to modulate the abundance of the transcription factor Phytochrome Interacting 4 (PIF4), the key regulator of petiole and hypocotyl elongation under shading and warm temperature[56,57]. We generated a hypoxia-stabilised version of PIF4 fused to the HIF$_{ODD}$ peptide. To minimise interference from other signalling pathways, we selectively mutated specific PIF4 residues whose phosphorylation in response to light, temperature and hormones is known to stimulate proteasomal

degradation[58] (Fig. 5A). Therefore, we expected this chimeric PIF4 version to be specifically controlled by hypoxia, due to HIF$_{ODD}$-dependent proteolysis. We sequentially transformed Col-0 plants with two different T-DNAs, one carrying both the *SLY1-2-VHL* and *pUBQ:HIF$_{ODD}$-PIF4* cassettes, and a second to express *PHD3*, according to the strategy adopted before (Supplementary Figs. 5B, 7A). *HIF$_{ODD}$-PIF4* over-expression with a *UBQ10* promoter caused an elongated petiole phenotype that could not be inhibited by PHD3, despite the use of a strong *35S* promoter (Supplementary Fig. 13). We therefore resorted to a moderately active ubiquitous promoter, from the *ROTUNDA3 (RON3, At4g24500)* gene, to control the *HIF$_{ODD}$-PIF4* transgene. We selected two independent lines for which we could segregate *PHD3* out (Supplementary Fig. 14A) and confirmed that hypoxia (1% $O_2$, 6 h) was able to increase HIF$_{ODD}$-PIF4 protein but not mRNA (Fig. 5B, C, Supplementary Fig. 14B, C). These plants exhibited slower leaf production and accelerated flowering (Supplementary Fig. 14D, E). The latter phenotype could result from increased PIF4 activity, reported to promote reproductive transition in Arabidopsis[59], in the chronically hypoxic shoot apical meristem[32]. Expression of *pRON3:PIF4-HIF$_{ODD}$* significantly stimulated growth of both hypocotyl and petiole length, a phenotype that was reversed, at least in part, by co-expression of *PHD3* (Supplementary Fig. 14F–H). This indicated that our strategy to adopt PHD/VHL-dependent degradation of

developmental regulators is successful at controlling organ elongation. Finally, we tested whether environmental hypoxia could relieve the repression imposed by PHD3 and SLY1-2-VHL on the chimeric $HIF_{ODD}$-PIF4 (Fig. 5D). We submerged 15-day-old plants in distilled water and measured petiole growth after 4 days (Fig. 5E, F). The wild type exhibited moderate elongation, which was insufficient to push the leaves near the water surface (Fig. 5E). Under the same water column, petioles of the transgenic plants grew twice as much, bringing blades closer to the atmosphere (Fig. 5E, F). This result confirmed the possibility of adopting the PDH/VHL-based hypoxia sensing system to control growth in response to changes in ambient $O_2$ availability.

## Discussion

With this work, we engineered an orthogonal hypoxia-responsive system for plant cells inspired by the $O_2$-dependent degradation of the mammalian HIF proteins (Fig. 1B). We showed that this system is applicable to control stability, and thus activity, of different proteins, resulting in the possibility to modulate different outputs in an $O_2$-sensing system. This approach aligns with those aimed to orthogonally control protein stability in mammalian and plant cells[60–62]. Successful engineering of selective proteolysis has been achieved with proteolysis- and autophagy-targeting chimeras[63,64], strategies that hold great potential for therapeutic applications and, more recently, also for agricultural applications[65]. Our exercise diverges from these in its aim to connect the output to a stimulus, cell $O_2$ concentration, which is affected by both endogenous and external factors.

The process of design, test and optimisation of the components required for this molecular device to perform the desired function highlighted old and revealed new opportunities and limitations of applying the synthetic biology framework to complex multicellular organisms. First, we confirmed that single-cell systems, such as transiently transformed protoplasts, can be effectively used to test the performance of genetic circuits (Figs. 1C–F, 2A-C), before successful export to stable transformation in whole plants (Figs. 3–5). For example, our system performed similarly in both contexts, when it was used to control the stability of luciferase reporters (Supplementary Fig. 2B, Fig. 2D). Differences in output dynamic range seemed to be associated with the kind of protein used as reporter, as highlighted in our applications of the system to generate low $O_2$ reporters based on luminescence or fluorescence (Fig. 2B, D, F).

One of the main challenges faced in this exercise of biological engineering was the difficulty to estimate, and account for, the heterogeneous behaviour of each component in a multicellular system. For example, cis-acting regulatory elements, such as promoters and regulatory mRNA untranslated regions, are strongly affected by their context, such as the chromatin state, which can differ greatly between cell types. As one of the few attempts to implement multi-components genetic circuits in plants[66–68], it was interesting to observe the strong effect that alternative elements produced on the output (Supplementary Fig. 5C). This consideration becomes even more relevant when the genetic circuit relies on the control of protein stability, a process for which the characterisation of components is more limited than transcription and translation. Such a limitation highlights the need for the community to produce a quantitative characterisation of degradation determinants. Despite these constraints, the success of our strategy paves the way for further optimisation and implementation. Our attempts to alter output dynamics using structure-inspired PHD mutations that impact the catalytic site and neighbouring residues did not succeed (Supplementary Fig. 6D, E), highlighting the difficulties in increasing the sensitivity of dioxygenases to hypoxia while maintaining high activity under aerobic conditions. The PHDs' gas tunnel is a promising alternative target to test in the future, based on the enhanced hydroxylation efficiency achieved in mammalian cells[69].

Swapping of $O_2$-sensing mechanisms between animal and plant cells (Fig. 3) is an exercise that interfaces with both fundamental and applied research. From a fundamental perspective, it allows experimental testing of the evolution of $O_2$ sensing in eukaryotes and its diversification between plants and animals. Intriguingly, both kingdoms adopted proteostatic control of transcription factors, on top of which additional regulatory mechanisms exist, to reprogramme cells to cope with reduced oxygen availability. While it has been speculated that this is a requirement and the optimal solution for the multicellular organisation that involves stem cells[18], it is puzzling that animals and plants diverged so much when their last common ancestor likely contained the biochemical functions to perform both N-cys-degron and VHL-dependent degradation. Successful control of transcriptional responses rewiring endogenous transcriptional regulators to PHD-primed degradation in Arabidopsis allows us to exclude intrinsic metabolic of signalling interference that would prevent this mechanism from working.

Animal cells control responses to hypoxia via the Cys branch of the N-degron pathway[11], although none of the N-Cys degron substrates characterised so far are transcription factors. This enables regulation of cellular processes occurring outside the nucleus, such as at the plasma membrane, in the cytoplasm, and in other cellular compartments[70–72]. It is tempting to speculate that this pathway also provides a faster response system to fluctuating $O_2$ levels than the HIF-based pathway, as it directly modulates the activity of regulatory proteins rather than requiring transcription and translation to elicit a response. In contrast, evidence for the involvement of PHD activity in regulating protein stability in plant cells is lacking, although less direct roles for these enzymes in plant responses to hypoxia have been proposed[73].

Differences in the regulation of transcriptional responses to hypoxia in plant and animal cells may also derive from the distinct ways in which cells in these organisms experience oxygen limitation. While chronic hypoxia of specific cell types, including stem cells, is characteristic of both systems[74,75], acute hypoxia is typically more deleterious in metabolically active animals, where ischaemia and subsequent reoxygenation cause rapid cellular damage[76,77]. In many metazoans, oxygen delivery through circulatory systems helps meet tissue oxygen requirements. Plants, by contrast, lack active oxygen transport and are generally more resilient to hypoxia, particularly during short-term exposure[78]. Moreover, plant cells are more likely to experience hyperoxia as a result of photosynthetic oxygen production[76,79]. The combination of these metabolic and anatomical constraints has likely played a major role in shaping the architecture of oxygen-sensing systems in the two kingdoms.

This study also revealed the existence of additional regulation exerted on RAP2.12, which determines the transiency of its accumulation in hypoxia, and its increase in the dark (Fig. 3B, C, Supplementary Fig. 8A, B). Similar, although slower, dynamics have been reported for the ERFVII RAP2.3[30]. Remarkably, target gene expression correlated with the protein abundance of native and synthetic RAP2.12 at early timepoints in the comparison between aerobic and hypoxic plants. This was not observed at later time points, which is likely explained by additional post-translational modifications, including phosphorylation that controls RAP2.12's nuclear localisation and transactivation capacity[80,81]. N-degron-dependent control of HIF stability, upstream of PHD regulation, has been proposed recently[82]. While it has been shown that N-cysteine oxidation determines substrate stability with similar $O_2$-sensitivity to prolyl-hydroxylation in animal cells[83], our swap exercise of $O_2$-sensing system between plant and animal cells hinted at higher proteolytic efficacy for the N-degron pathway when compared with VHL-promoted degradation. It is tempting to speculate that this might reflect the need to switch on adaptive responses at different $O_2$ levels.

In summary, we successfully generated a PHD- and $O_2$-dependent degradation of selected proteins through the engineering of a chimeric F-box protein. The evolutionary question of why animals and

plants rely on different dioxygenases to control the stability of transcription factors that regulate hypoxia-responsive gene expression remains far from fully answered. Nevertheless, our results show that these systems are, in fact, functionally interchangeable across kingdoms, providing a foundation for future efforts in rational design or directed evolution aimed at matching gene expression dynamics to those governed by endogenous pathways.

By adding the $HIF_{ODD}$ fragment to different reporters and effector modules, we demonstrated the possibility to monitor cell oxic states as well as to drive advantageous developmental traits for submergence tolerance. From an applied perspective, wiring the synthetic $O_2$-sensing system to control petiole elongation in Arabidopsis (Fig. 5) pioneers, together with other studies[84], the engineering of adaptive responses to environmental cues. While it remains to be established whether promotion of fast petiole elongation can be useful in specific agroecosystems, it is possible to connect the novel $O_2$-sensing mechanism to repressors of petiole elongation, thus favouring the quiescent syndrome typical of flash flood-tolerant species. The highly transgenic nature of the system prevents its immediate application in agriculture in many countries; nevertheless, it demonstrates feasibility and potential for synthetic biology to improve crops' ability to cope with abiotic stresses, including those exacerbated by climate change. While we here conveniently used Arabidopsis as a testbed, given the compatibility of molecular components and the availability of simple transformation and selection protocols, the chimeric $O_2$-sensing system is likely exportable to many other crops, including monocots.

## Methods

### Plant materials and growth conditions

The *Arabidopsis thaliana* Columbia-0 (Col-0) ecotype was used as the wild-type background in all the experiments. The *erfVII* mutant[30], which carries T-DNA insertions in the genes *At1g72360*, *At2g47520*, *At1g53910*, *At3g14230* and *At3g16770*, was provided by Michael J. Holdsworth. Plants were either grown in soil on a peat:perlite 3:1 mixture, or in sterility on agarized (8 g l$^{-1}$) half-strength Murashige and Skoog (MS) medium supplemented with 1% sucrose, after sterilisation with 70% ethanol and 10% commercial bleach solution and 5 washes with sterile distilled water. In both cases, seeds were vernalized at 4 °C for 48 h in the dark and then germinated at 22 °C day/18 °C night with a photoperiod of 12 h. The plant material was subjected to short-duration low-oxygen treatments as specified in the text and figure legends by flushing a mixture of pure nitrogen ($N_2$) and compressed air to reach the desired $O_2$ concentration inside hermetic boxes. For treatments longer than 24 h, we instead used a Hypoxic Workstation (Whitley). For submergence treatments, plants grown in sterility in magenta boxes were submerged with water at an approximate depth of 3 cm. The treatment began at the end of the day for a period of 72 h.

### Generation of transgenic plants

*A. thaliana* plants were transformed with *Agrobacterium tumefaciens*, applying the floral-dip method[85]. Selection of transformed plants was carried out using appropriate antibiotics, herbicides or fluorescent seed selection. Transgene presence was assessed by PCR.

### In silico protein complex assembly

The structure of the Cul1-Rbx1-Ask1-TIR1 complex was predicted based on atomic coordinates from the Protein Data Bank (PDB) entries 2P1Q and 2P1O (10.1038/nature05731), which include the TIR1 and ASK1 protein structures, and AlphaFold[86] entries AF-Q94AH6-F1-v4 and AF-Q940X7-F1-v4, representing the Arabidopsis Cullin1 and Rbx1 structures, respectively.

Similarly, the Cul1-Rbx1-Ask1-SLY1 complex structure was predicted using the PDB entry 2P1Q containing the Ask1 protein structure, alongside AlphaFold entries AF-Q94AH6-F1-v4, AF-Q940X7-F1-v4 and AF-Q9STX3-F1-v4, which correspond to Cul1, Rbx1 and SLY1,

respectively. AlphaFold models AF-Q94AH6-F1-v4, AF-Q940X7-F1-v4 and AF-Q9STX3-F1-v4 were retrieved from the AlphaFold protein structure database. Structural predictions of the Cul1-Rbx1-Ask1 complex with TIR1 and SLY1 were carried out using the full Cul1–Rbx1–Skp1–Skp2 complex as template (PDB: 1LDJ[87]).

The structure of the SCF complex Cul2-Rbx1-EloB-EloC-VHL-$HIF_{ODD}$ was predicted using atomic coordinates 1LQB[88] and 1VCB[86], which include ELOB, ELOC, VHL and $HIF_{ODD}$, and 5N4W[89], which provides the structure for VHL, ELOB, ELOC, Cul2 and Rbx1. Structural alignment and figure rendering were performed using VMD and PyMol software.

### DNA cloning and assembly

Transcriptional units were either cloned from genomic DNA, cDNA or de novo synthesised by GeneArt (Thermo-Fisher Scientific). Protein-coding sequences were codon optimised using the EMBOSS Backtranseq online tools[90]. A full list of synthetic sequences and plasmids used in this work is provided in Supplementary Tables 1, 2, respectively. All maps of the plasmids used in this work are provided as Supplementary Data 3. DNA fragments provided with a 5′-CACC additional sequence or flanked by restriction sites were initially subcloned either in the pENTR/D-TOPO® vector, to be recombined into destination vectors using Gateway™ LR Clonase™ II Enzyme mix (catalogue number 11791020, Thermo-Fisher Scientific), or in the pCR™2.1-TOPO®-TA vector (Thermo-Fisher Scientific), to be inserted into expression vectors via a restriction-ligation strategy, respectively. Plasmid maps were generated with Serial Cloner (Serial basic).

For protoplast transformation, the 35S:FboxTIR1-VHL, 35S:$HIF_{ODD}$-GAL4DBD-RAP2.12AD-3xFLAG, 35S:SLY1-VHL, 35S:SLY1-2-VHL and 35S:FboxSLY1-10-VHL plasmids were generated from synthetic coding sequences, purchased as DNA strings from GeneArt (Thermo-Fisher Scientific), further ligated in the pENTR/D-TOPO® vector (Thermo-Fisher Scientific) and recombined into the p2GW7[91] vector via Gateway™ cloning. The 4xUAS:Fluc plasmid has been described in Bui et al. (2015). The 35S:PHD3 plasmid has been described in Iacopino et al. (2019)[24]

The $O_2$ratio construct was produced by overlapping PCR. A 5′-terminal 3xHA-Renilla-UBQ fragment was amplified with the primers 3xHA-Fw and UBQ-Rv using the C-DLOR plasmid[92] as a template, while the fragment bearing the Firefly Luciferase CDS fused to the $HIF_{ODD}$ was amplified with the primers UBQ_Luc-Fw and Luc_HIF_Rv using the 4XUAS:Fluc plasmid[38] as a template. The Luc_HIF_Rv primer was designed to include the full $HIF_{ODD}$ coding sequence. The two fragments were joined together with a second PCR using the oligonucleotides 3xHA-Fw and Luc_HIF_Rv. The resulting PCR product was cloned into the pENTR/D-TOPO®. The construct 35S:$O_2$Ratio/pUB-Q:SLY1,2-VHL used for Arabidopsis infiltration was generated in three steps. First, a synthetic construct consisting of the Arabidopsis UBQ10 promoter[93], flanked by the SacI and AfeI restriction sites, and followed by the 35S CaMV terminator[94], flanked by EcoRI and a further SacI restriction site, was ligated into the pK7WG2[91] plasmid using the compatible SacI restriction site. Then, the SLY1-2 coding sequence was cloned in between the UBQ10 promoter and 35S CaM terminator using the AfeI and EcoRI restriction sites, to obtain a destination plasmid named pK7WG2/pUBQ10:SLY1-2. Finally, this destination was recombined with the entry plasmid containing $O_2$ratio via Gateway™ cloning.

The NLS-$O_2$ratio and $O_2$ratio$^2$ variants were generated by restriction/ligation cloning operated on the $O_2$ratio entry vector. SacI and XbaI were used to remove a portion of the UBQ10 and FLuc sequence, and a DNA fragment designed to contain the excised regions and the additional $HIF_{ODD}$ or NLS element was ligated using the same restriction sites. The nano$O_2$ratio$^2$ entry vector was synthesised by GeneArt (Thermo Fisher Scientific). The nano$O_2$ratio$^2$ variants were generated by digesting the nano$O_2$ratio$^2$ entry vector using the SacI and XhoI restriction sites and ligating an identical DNA sequence devoid of the

first HIF-COOD. To replace the CaMV 35S Omega leader with the 5′UTR region of the Arabidopsis *ADH1* gene, pK7WG2/pUBQ10:SLY1-2 was cut with StuI and SpeI and ligated with a synthetic DNA string containing the excised region of the 35S promoter followed by *ADH1* 5′-UTR, flanked by compatible restriction sites, thereby turning the original vector into the new pK7WG35SUTR$^{ADH}$/pUBQ10: SLY1-2 Gateway destination vector.

All O$_2$ratio variants entry vectors were then recombined using Gateway™ cloning in these two destination vectors to generate the binary expression vectors reported in Supplementary Fig. 5.

The NLS-VENUS-HIF$_{ODD}$ construct was purchased as a DNA string from Geneart (Thermo-Fisher Scientific) and ligated into the pENTR/D-TOPO®. The p16:NLS-VENUS-HIF$_{ODD}$/pUBQ: SLY1-2-VHL plasmid was generated by recombination of the NLS-VENUS-HIF$_{ODD}$ entry with pK7WG2/pUBQ10:SLY1-2 and subsequent replacement of the 35S promoter. Specifically, the expression plasmid was cut with StuI and SacI restriction enzymes and ligated with a compatible p16 promoter sequence[45] amplified using primers p16_Fw and p16_Rv and from a p16-FLIPnls43 template vector[95].

The construct pRAP2.12:HIF$_{ODD}$-2xHA-D13RAP2.12/pUBQ:SLY1-2-VHL was generated as follows: a synthetic DNA string containing the chimeric HIF$_{ODD}$-2xHA-D13RAP2.12 coding sequence was ligated in pENTR/D-TOPO® and recombined with the pK7WGpUBQ10:SLY1-2 destination vector. A 2kb-long upstream region of the *RAP2.12* promoter was amplified using NcoI_pRAP_Fw and NcoI_pRAP_Rv from Arabidopsis genomic DNA. The 35S promoter was then removed from the expression plasmid using NcoI restriction enzyme and substituted with the *RAP2.12* promoter fragment via ligation. Finally, the kanamycin plant selection marker was removed using ApaI and CpoI restriction enzymes, to be replaced with a fluorescent selection cassette. The pOLE_Fw and mCHERRY_Rv primers were used with a pKIR1.0[96] template plasmid to amplify an expression cassette consisting of the Arabidopsis oleosin promoter (pOLE1) and a downstream fusion sequence between the red fluorescent protein mKATE2 and the oleosin gene (*OLE1*). This fluorescent cassette, flanked by compatible ApaI and CpoI sites, was subsequently ligated with the backbone obtained before. The construct pRAP2.12:RAP2.12-2xHA was synthesised in an entry plasmid by GeneArt (Thermo-Fisher Scientific) and subsequently recombined into the pH7WG[91] binary vector via Gateway™ cloning.

To generate the pPCO4:PHD3 expression vector, the PHD3 entry plasmid was recombined Gateway™ cloning into the plasmid pH7WG-promPCO4[46]. The expression plasmid was digested with ApaI and CpoI, and an expression cassette consisting of the Arabidopsis At2S3 promoter driving GFP expression was amplified from pFP91[97] with primers At2S3_Fw and t35S_Rv and subsequently ligated. PHD3 mutations were obtained by site-directed mutagenesis with PHD3Asp137His Fw and Rv, PHD3Asp137Glu Fw and Rv and PHD3Arg205Lys Fw and Rv. The HIF$_{ODD}$ -PIF4 sequence were purchased as a DNA string and ligated into the pENTR/D-TOPO®. The pSIC:PIF4-HIF$_{ODD}$/pUBQ:SLY1-2-VHL plasmid was obtained by recombining the PIF4-HIS entry into the pK7WG2/pUBQ:SLY1-2 and replacing the 35S CaMV promoter using StuI and SpeI restriction sites. The SIC/RON3 promoter was amplified by PCR using primers pAt4g24500_Fw and pAt4g24500_Rv using Col-0 genomic DNA as template and then ligated using compatible restriction sites.

### Hypoxia, submergence, and bortezomib treatments
For hypoxia treatments, 7-day-old in vitro−grown seedlings were exposed to hypoxia (specified in each figure legend as % O$_2$/N$_2$ [v/v]) for 6 h in the dark (starting at 8 a.m.), while control plants were kept in the dark under atmospheric conditions (21% O$_2$). For Western blot or qRT−PCR analyses, each biological replicate consisted of at least five seedlings. For submergence tolerance assays, 14-day-old seedlings grown in Magenta boxes were submerged in 60 mL of distilled water

for 4 d in the dark. Control plants were kept in the dark without submergence. Bortezomib (CAS 179324-69-7; Santa Cruz Biotechnology) treatments were conducted on 7-day-old Arabidopsis seedlings transferred to six-well plates containing fresh half-strength MS medium supplemented with 50 μM BZ or 0.02% DMSO (mock control) for 3 h.

### Protoplasts transformation and transactivation assay
Protoplasts isolation and transformation were performed as reported by Yoo et al.,[98] with the modifications described in Iacopino et al.[24]. Firefly and Renilla luciferase activity was measured using the Dual-Luciferase® Reporter Assay System (E1910, Promega), Nanoluciferase and Firefly luciferase activity was measured using Nano-Glo® Dual-Luciferase® Reporter Assay System (N1610, Promega). Relative luciferase activity for O2ratio variants was calculated as the ratio between the HIF$_{ODD}$-lin ked luciferase and the HIF$_{ODD}$-independent luciferase.

### Phenotypic analyses
Petiole and hypocotyl lengths were measured from images using ImageJ software[99]. Petioles were pressed between two layers of Scotch tape and scanned with an EPSON Perfection V750 PRO scanner at a 400 DPI resolution. Hypocotyl images were captured using a Nikon Coolpix P520 digital camera. Root lengths were measured from images scanned from plastic square plates using ImageJ software.

### Human cell culture and hypoxia exposure
HKC-8 wild type and DKO cells were cultured in Dulbecco's Modified Eagle Medium, supplemented with 10% foetal bovine serum, 2mM L-Glutamine, 100 U/ml penicillin and 10 μg/ml streptomycin. Cells were maintained at 37 °C under an atmosphere of 5% CO2 in air. Hypoxic incubations were conducted in an atmosphere-regulated workstation set at 1% O2: 5% CO2: balance N2 (Invivo 400, Baker-Ruskinn Technologies).

### Genomic DNA extraction from DKO cells
Genomic DNA was extracted from cell pellets by incubation in lysis buffer (100 mM Tris pH 8.0, 5 mM EDTA, 200 mM NaCl, 0.2% SDS and 100 μg/ml Proteinase K) with 18 shaking at 55 °C for 2 h, followed by isopropanol precipitation and 70% ethanol wash. Precipitated genomic DNA was re-suspended in water. Genomic target regions were PCR amplified with gHIF1/2α primers mentioned in Supplementary Table 1 using the PrimeSTAR® GXL DNA Polymerase according to the manufacturer's protocol, using 30 cycles, 98 °C denaturation temperature and 60 °C annealing temperature. PCR products were separated using agarose gel electrophoresis. DNA was then purified with QIAquick® PCR Purification Kit (Qiagen), and Sanger sequenced.

### Luciferase assays from human cell cultures
Luciferase assays were conducted using the Dual Luciferase Reporter kit (Promega) and measurements taken using the Fluostar® Omega Microplate Reader. In experiments that did not include RnLuc expression, the step involving the Stop & Glo Reagent was omitted. Human cells grown in 12-well plates were co-transfected with 500 ng of PGL3 PGK6 TkPluc and 100 ng of PGL4 RL plasmid DNA[52]. 6−8 h after transfection, cells were incubated in normoxia (21% O$_2$) or hypoxia (1% O$_2$) for 16−18 h. After the incubation period, cells were washed in phosphate-buffered saline, lysed with 100 μl 1X Passive lysis buffer and incubated at -80 °C for at least 2 h. Homogenised Arabidopsis tissues and protoplasts were similarly lysed with 100 μl 1X Passive lysis buffer. Prior to measurement, cell lysates were spun at max speed at room temperature for 2 min using a benchtop centrifuge. Firefly luciferase activity was assayed by combining 6 μl cell lysate with 30 μl Luciferase Assay Reagent II (or 10 μl cell lysate + 50 μl LAR II). Renilla luciferase was measured by adding an equal volume of the Stop & Glo Reagent.

## Generation of stable cell lines

HKC-8 DKO cells were transfected with the pcDNA3 plasmid carrying HIF1 α -, HIF1 α PPAA, RGS41-11-HIF1α PPAA using GeneJuice® (Novagen). After 24 h, cells were seeded at limiting dilution in selective medium containing 0.8 mg/ml G418. Resistant clones were isolated and expanded. Clones stably expressing HIF1α proteins were identified by transfection with PGL3 PGK6 TkPluc and PGL4 RL vectors and luciferase assay, as described above, to detect HIF1α transcriptional activity.

## Confocal imaging

Confocal investigations were performed using a Zeiss LSM800 confocal microscope. For sub-cellular localisation studies, GFP fluorescence was excited with 488 nm laser light and collected with a 497–554 nm long-pass emission filter. Chlorophyll autofluorescence was excited at 633 nm and collected at 650–750 nm. Nuclei were stained with 1 μg μl$^{-1}$ 4′,6-diamidino-2-phenylidone (DAPI, Sigma-Aldrich), and fluorescence was excited at 405 nm and collected at 410–470 nm. For Venus visualisation in roots, plants were stained with 10 μg μl$^{-1}$ propidium iodide (PIE) (Sigma-Aldrich) cell wall stain. The roots were observed with a 20x objective lens under ZEISS LSM 800 Laser Confocal Scanning Microscope. HIF-NLS-Venus seedlings were fixed with 4% paraformaldehyde and stained with ClearSee[100] supplemented with 1 μl/ml SCRI Renaissance 2200 (SR2200). Venus fluorescence was excited with 488 nm laser light and collected with a 520–560 nm. PIE was excited with 488 nm laser light and collected at 650–700 nm. Images were analyzed with the ZEN 2010 software (Zeiss).

## SDS-PAGE and immunoblotting

Total Arabidopsis proteins were isolated from 1-week old seedlings grown on vertical plates using a buffer containing 50 mM Tris (pH 7.5), 0.1% w/v SDS and Protease Inhibitor Cocktail (cOmplete™, Mini, EDTA-free Protease Inhibitor Cocktail, Sigma-Aldrich; 11836170001). From leaf mesophyll protoplasts, total proteins were instead extracted using a buffer containing Laemmli and 5% v/v β-mercaptoethanol. Equal total protein amount (70 μg) was resolved by SDS-PAGE and then transferred to PVPF membrane using MiniTrans-Blot electrophoretic transfer cell (Bio-Rad). To detect the HA tag, membranes were probed with anti-HA primary antibody (Sigma-Aldrich; H3663) at 1:2000 dilution. Membranes were then probed with HRP-conjugated anti-mouse secondary antibody (Sigma-Aldrich; 12–349) at 1:10,000 dilution. To detect the FLAG-tagged GAL4 activation domain, membranes were probed with the monoclonal HRP-conjugated, anti-FLAG antibody (A8592, Sigma-Aldrich) at 1:1000 dilution. Immunoblots were developed with SuperSignal™ West Pico PLUS Chemiluminescent Substrate (Thermo Fisher Scientific) using the iBright CL1500 Imaging System (Thermo Fisher Scientific). To detect HIF$_{ODD}$-PIF4, a recombinant anti-PIF4 antibody was used (AS163955, 2B Scientific) at 1:1000 dilution. Membranes were then probed with HRP-conjugated anti-goat secondary antibody (AS09 605, Agrisera) at 1:10,000 dilution.

For immunoblot analyses of human cells, -1 × 106 cells were washed in phosphate-buffered saline, lysed in SDS lysis buffer (50 mM Tris, pH 6.8, 2% SDS, 10% Glycerol), sonicated, and mixed with Laemmli 6X SDS-PAGE sample buffer. Proteins were then separated using a 9% SDS-PAGE gel, transferred to polyvinylidene difluoride membrane (Immobilon-P, Millipore) and blocked in 4% fat free milk (in phosphate-buffered saline containing 0.1% Tween 20). Primary antibodies used were: Purified Mouse Anti-Human HIF-1α (610959, BD Transduction Laboratories™) and anti-HIF-2α (mouse monoclonal, 190b). HRP-conjugated secondary antibodies (DAKO) and chemiluminescence substrate (West Dura, 34076, Thermo Fisher Scientific) were used to visualise proteins, using the ChemiDoc XRS+ imaging system (BioRad). After immunoblot imaging, membranes were stained with Coomassie brilliant blue to visualise total protein, as a reference for sample loading.

## Plant RNA isolation and qPCR analyses

Total RNA was extracted from 100 mg of frozen-ground seedlings as described previously[33]. One microgram of total RNA was treated with DNase (Thermo Scientific™) to remove genomic DNA before retro-transcription using qPCRBIO cDNA Synthesis Kit (PCR Biosystems). Real-time quantitative PCRs were performed in 10 μl volume using a 2X Power SYBR™ Master Mix (Thermo Fisher Scientific™), 10 ng cDNA and 0.2 μM of specific reverse and complement primers for each gene to be tested. Thermal cycling and fluorescence acquisition were carried out with the StepOnePlus™ Real-Time PCR System. Ubiquitin10 (*AT4G0532*) was used as a housekeeping gene for Arabidopsis analysis. A full list of the primers used for qPCR is included in Supplementary Table 1. Relative expression of each individual gene was calculated using the $2^{-\Delta\Delta Ct}$ method[101].

## RNA sequencing

RNA was isolated using GeneJET RNA Purification Kit (Thermo Scientific™) as per the manufacturer's instructions. Genomic DNA and cDNA library preparation were carried out as described in Dalle Carbonare et al.[48]. Libraries were sequenced on an Illumina NovaSeq 6000 platform. Reads cleaning, mapping and counting were performed as described in Dalle Carbonare et al.[48], and differential expression analysis was performed using the DESeq2 R package (v1.20.0), with *p*-values adjusted by the Benjamini-Hochberg method to control for FDR.

## Statistical analysis

Ordinary two-way and one-way analysis of variance (ANOVA) and multiple comparisons for statistical differences were performed with R or GraphPad Prism 9 for Windows 10.

## Mathematical modelling of RAP2.12/PCO4 and HIF-RAP2.12/PHD kinetics

Model description and assumptions. The two models that were developed to describe the expression dynamics of Hypoxia Responsive Genes (HRG) in either the PHD/VHL- or the N-degron pathway-based oxygen sensing system consist of two coupled Ordinary Differential Equations (ODEs). Within these equations, we incorporated both Michaelis-Menten Kinetics and Mass Action laws to describe the gene expression and biochemical reaction network processes (Supplementary Tables 3, 4)[102]. The overall processes involving RAP2.12 expression, degradation, and downstream HRGs activation were illustrated in Supplementary Fig. S10A.

In our models, the RAP2.12 protein is synthesised at a rate $k_1$ and degraded at a rate $k_2$. In the *promRAP2.12:RAP2.12-2xHA* scenario, where RAP2.12 is the only ERFVII expressed in plant cells, the N-Cys-degron pathway promotes $O_2$-dependent RAP2.12 degradation following Michaelis-Menten kinetics, with catalytic constant $k_3$, and Michaelis constants $Km_1$ and $Km_2$. In this system, HRG expression, which in our model is assumed to be exclusively driven by RAP2.12, follows Michaelis-Menten kinetics at a rate represented by $k_4$ and $Km_3$ and decay occurring at a rate of $k_5$. In the HIF$_{ODD}$-RAP2.12 system, we applied the same parameters for RAP2.12 protein synthesis and degradation as well as for HRG activation. However, in this chimeric system, HIF$_{ODD}$-RAP2.12 undergoes $O_2$-dependent degradation through enzymatic prolyl-hydroxylation, characterised by rates $k_6$, $Km_4$, and $Km_5$. Since the production and degradation rates of RAP2.12 are unknown, we used the HIFα protein production and degradation rates from the literature. Similarly, the PHD concentration was set to 0.1 μM based on published models[103]. These assumptions do not affect the validity of our conclusions.

To simplify our model, we assume that all reactions occur in a homogeneous environment, without accounting for compartmentalisation or specific microenvironments within the cell. We further assume that each reaction species—such as RAP2.12, PCO, PHD, $O_2$, and mRNA—is evenly distributed throughout the system. To focus on the oxidation and hydroxylation processes, we simplify the model by assuming that RAP2.12 and $HIF_{ODD}$-RAP2.12 are degraded instantly following N-Cys and Pro oxidation, respectively.

Additionally, for comparability, we assume identical concentrations of PCO and PHD and at steady state under normoxic conditions that characterise the initial timepoint in the simulation. These enzyme concentrations are assumed to remain constant after exposure to 1% hypoxia. In a short timeframe, this is not unrealistic since protein synthesis is strongly reduced under hypoxia[104]. By maintaining the same levels of PCO and PHD, we can compare the dynamic hypoxic responses based solely on their catalytic activities. Accordingly, the model was designed as a constrained, hypothesis-generating framework that emphasised relative system behaviour under standardised conditions rather than absolute quantitative prediction across experimental platforms.

Non-linear fitting and simulation. We used MATLAB's 'Welsh' and 'Fair' weight functions within the nlinfit function. The rate constants $k_3$, $Km_1$, and $Km_2$ were re-fitted for the PCO-mediated degradation of RAP2.12, while $k_6$, $Km_4$, and $Km_5$ were re-fitted for the PHD-mediated degradation of $HIF_{ODD}$-RAP2.12. This three-dimensional nonlinear fitting, which is depicted in Supplementary Fig. S10B, was performed based on published kinetic data[22,44]. All parameters are listed in Supplementary Table 5.

To simulate hypoxic responses, the initial concentrations of RAP2.12/$HIF_{ODD}$-RAP2.12 and mRNA as inputs were obtained from steady-state simulations conducted under normoxic conditions (21% $O_2$) for 6 h (Supplementary Table 4). Subsequently, dynamic simulations of hypoxic responses (Fig. 4A) were conducted under 1% $O_2$ conditions for 4 h. Steady-state oxygen–response curves (Fig. 4B) were generated by simulating the model at a range of fixed oxygen concentrations (0.1–21%). For each oxygen level, the system was integrated for 6 h to reach steady state, and the simulated HRG expression was recorded.

## Reporting summary
Further information on research design is available in the Nature Portfolio Reporting Summary linked to this article.

## Data availability
Raw RNA sequencing data from this study are available in the Gene Expression Omnibus (GEO) under accession number GSE315308. Uncropped western blot images are available in Zenodo (https://doi.org/10.5281/zenodo.18074079). All additional data supporting the findings of this study that are not included in the article, Supplementary Information or Source Data are available from the corresponding authors upon request for non-commercial academic research; requests will be evaluated within 4 weeks and, if approved, data will be shared via a secure file-transfer link. Source data are provided with this paper.

## Code availability
The equations used to generate the results reported in this study are available in the materials and methods and supplementary information sections. The code is available at the link: https://github.com/ymiplant/orthogonal-hypoxia-responses-plants.git.

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

## Acknowledgements

We thank Dr. Norma Masson for training A.D.C. in human cell handling and western blotting. European Research Council Grant 101001320 'Synoxys' and Italian Ministry of University and Research PRIN grant 20173EWRT9 (FL); Biotechnology and Biological Sciences Research Council Interdisciplinary Doctoral Programme (BB/T008784/1, YH) and BB/Z516946/1 (FL, VS). Oxford University Press John Fell Fund 0009776 (FL, SI).

## Author contributions

Conceptualisation: V.S., S.I., B.G. and F.L. Methodology: V.S., S.I., L.D.C., A.D.C., Y.H., T.P.K., A.P. and M.N.T. Investigation: V.S., S.I., L.D.C., A.D.C., Y.H. and T.P.K. Visualisation: S.I. and V.S. Funding acquisition: F.L. and S.I. Project administration: F.L. and B.G. Supervision: F.L., B.G. and A.P. Writing – original draft: S.I., V.S., F.L., and B.G. Writing – review & editing: T.P.K. and A.P.

## Competing interests

The authors declare no competing interests.

## Additional information

**Peerreview information** *Nature Communications* thanks Zhixing Cao, who co-reviewed with Yiling Wang; Daniel Gibbs and the other anonymous reviewer(s) for their contribution to the peer review of this work. A peer review file is available.

