## [Transparent Peer Review file · Nature Communications]

Prolyl hydroxylase-dependent proteolysis enables orthogonal hypoxia responses in plants

Corresponding Author: Professor Francesco Licausi

Version 0:

Reviewer comments:

Reviewer #1

(Remarks to the Author)

This intriguing manuscript seeks to understand the evolution of O₂ sensing mechanisms in animals and plants through a synthetic biology approach.

Plants and animals both sense and signal oxygen availability through oxygen-induced degradation of master transcriptional regulators, but the precise molecular players differ. Plants employ cysteine oxidases which, under normoxia, tag ERFVII transcription factors for proteasomal degradation via the N-degron pathway; in animals, the alpha subunit of HIF is modified by PHD-mediated prolyl hydroxylation and consequently degraded via the von Hippel Landau E3 ligase (VHL). Under hypoxia, both ERFVII and HIF are stabilised and orchestrate distinct transcriptional responses.

The authors present an elegant series of engineering approaches informed by structural considerations to reconstruct the PHD-VHL system in Arabidopsis. The engineered O₂-dependent proteolytic system was able to drive expression of a reporter construct in leaf protoplasts and then deployed to create a ratiometric O₂ sensor using a dual luciferase system. This gave an O₂ readout in whole seedlings, albeit with a limited dynamic range. More usefully, the HIF oxygen dependent degron (HIFODD) was transferred to nuclear-targeted VENUS to provide cellular resolution. While this is an interesting proof of principle, the study lacks benchmarking against other methods to measure or report oxygen in plants (see: <https://doi.org/10.1093/plphys/kiae624>). Moreover, both these setups use reporters that require oxygen. This potentially constrains the use of these sensors but the HIFODD degron could in principle be fused to an O₂-independent FP such as UnaG.

The authors then go on to swap the N-cys oxidation and Pro hydroxylation pathways in plant and animal cells lacking the endogenous transcriptional regulators and demonstrate that both synthetic circuits are (to some extent) effective in controlling accumulation of RAP12 or HIF1- α . However, the N-degron pathway was more effective at removing both transcriptional regulators than the VHL pathway. Surprisingly, RAP2.12 bearing the HIF1 degron was much more stable in normoxia than hypoxia after 4 h (Fig. 3C and Fig. 8B) suggesting that protein abundance becomes uncoupled from O₂ availability. This remains to be explained; the authors speculate that other post-translational mechanisms operate and there is evidence for these in the literature, but it is not clear why they would differentially affect the synthetic RAP2.12 to destabilize it in hypoxia.

In line with the reporter experiments, modelling and measuring levels of hypoxia responsive transcripts demonstrated that the VHL pathway could also control physiologically relevant responses in plant cells, with a modest effect on plant survival and root growth under 1 % O₂. Finally, the authors correctly reason that engineering the endogenous O₂ sensing system of plants is likely to compromise plant fitness and therefore adopt the PHD-VHL system to control plant responses to submergence, using petiole elongation as an exemplar. This is an exciting result but is lacking information about any potential negative impacts on the plants (e.g. the pRON3:PIF4-HIFODD plants appear to have fewer leaves than the wild type) which would be relevant to translating this system to crops.

The manuscript is clearly and engagingly written, with attractive figures. I particularly appreciated the thought process evident in the manuscript and the honest reporting of approaches that had limited success. This sort of information is really helpful to others attempting similar projects. Whilst the work represents an impressive feat of synthetic biology, I am less convinced that the authors have fully answered the evolutionary question that they posed. This is evident in the Discussion

which focuses on interesting and pertinent issues around synthetic biology and only relatively briefly touches on the evolutionary question. A key point which is not given sufficient attention is that, in addition to the PHD-VHL-HIF system, animal cells also use the N-degron pathway to control protein stability in response to oxygen. This was discovered by the authors themselves: <https://doi.org/10.1126/science.aaw0112>) and is briefly alluded to in the manuscript. Presumably animal cells use the two systems in different contexts. The manuscript could be improved by considering this and also the different ways in which plants and animals encounter hypoxia.

Fig. 4 was the survival and root growth experiment performed more than once?

Was the O₂ concentration response of the PCO and SAD6 expression modelled?

“hypoxia-inducible version of PIF4” could imply that a hypoxia-responsive promoter was used. Perhaps rephrase to state “oxygen sensitive version of PIF4” or “hypoxia-stabilised version of PIF4”?

Fig. 5: were PIF4 protein levels quantified?

Reviewer #2

(Remarks to the Author)

Shukla et al present “Prolyl hydroxylase-dependent proteolysis enables orthogonal hypoxia¹ responses in plants”, where they test the transferability of oxygen sensing capabilities between vascular plants and metazoans. In plants, molecular oxygen is sensed using oxidation of a cysteine, forming a N terminal degron. Some metazoans use oxidation of proline, forming a degron that is recognized by VHL (HIFODD peptide). In this contribution, the authors tested the transferability of the metazoan pathway into oxygen-insensitive lines of *Arabidopsis*. The authors had previously established that the PHD enzyme can use molecular oxygen and selectively conjugate the HIFODD peptide in planta. In this contribution they created a chimeric VHL in a CRL E3 ligase complex. What worked was genetically fusing truncated SLY1 variants to the VHL beta domain. As shown in Figure 1D, this construct was successful in rescuing partial oxygen-dependent reporter gene expression in planta. They demonstrated that transgenic lines containing a fusion partner of the HIFODD peptide, PHD enzyme, and chimeric VHL could control molecular oxygen-dependent gene expression in a relative luciferase assay, using a ENUS reporter, and in the plant native RAP2.12 native system for sensing hypoxia in plants. To complete the story, they added plant-like oxygen regulation in mammalian cells by addition of a RGS41-11 N-degron in HIF with a double proline to alanine mutant. This addition by itself restored oxygen-dependent regulation of protein expression in this cell line.

The paper was a fun read and interesting to me from the sense of transferability of post-translational modifications for E3 ligase-mediated degradation across domains of life. The protein engineering was standard from a chemical biology perspective, but – to their credit - the authors showed what did and did not work in their system. It is a bit confusing that catalytic knockdowns of PHD dioxygenase did not change the overall system dynamics in planta, but I view this intriguing finding more in line with a follow-up story. I have the following minor comments:

1. For the Western blots reported, the loading controls on gels (Fig 1F, S1 except for the mammalian cell line experiment) appear spotty in some cases; it would benefit the paper to redo these experiments. The conclusions of expression at different environmental conditions and time points rest on these experiments. Additionally, the unmodified gels must be deposited in Zenodo or other DOI-linked resource for fair evaluation.
2. The definition of relative Fluc activity (Figure 2B) is not discussed in the methods or the main results section (unless I missed it). I'm confused because the line #1 shows a baseline of relative Fluc at 2, and I would have expected 1.0. Please clarify.
3. For the mathematical modeling section, I'm a bit surprised that the model showed high micromolar steady state concentrations of the transcription factor proteins. I couldn't tell from Supp Tables 3-5 as to whether the equations are dimensionally homogeneous; if not a unit conversion mistake, the authors should comment on the validity and limitations of their model considering the predicted concentrations.
4. Small – the UBG fusion protein is a clever approach. Most use variations of a 2A peptide. Is there a reference for this UBG fusion strategy?

Reviewer #3

(Remarks to the Author)

This interesting paper by Shukla et al investigates an alternative oxygen sensing scenario in plants – one where the predominant HIF based perception and transduction mechanism of animals is transferred and used to replace the endogenous N-degron pathway-based sensing system, as well as regulate new flooding-induced processes that could facilitate stress survival. The authors speculate nicely in the introduction that the kingdom specific differences in oxygen sensing (despite both being dependent on proteolysis) might relate to the different requirements of and degrees of hypoxia experienced by hetero- vs autotrophs. It is an interesting question, and overall, the authors provide solid evidence for the efficacy of their synthetic biology approach, which also offers new insights into the evolution of hypoxia sensing.

The molecular engineering utilised in this study is excellent. The iterative process of designing chimeric F-box-VHL proteins is impressive and shows that the authors carefully considered how best to design such a system and explored a variety of options to get the most potent construct. I like that they included the negative results obtained whilst trying to use the TIR1 F-

box, as this provides insight into the design-test-refine process that One goes through with this type of project. The combined flux activity and western blot measurements of HIFOOD-GAL4-AD convincingly show that they were able to generate a de novo engineered O₂ sensing mechanism in plant cells. The O₂ ratio (i.e. Luc1-Luc2) construct is also a very clever tool for allowing assessment of oxygen dependent stability of Luc2OOD using the Luc1-UBQ as a control for normalisation, and the authors clearly demonstrate the portability of their synthetic system for the oxygen dependent control of diverse reporters, including Luc and VENUS.

The experiments where RAP2.12 is uncoupled from N-degron regulation and instead diverted to VHL/PHD mediated degradation are well described. It is interesting that HIFOOD-RAP2.12 is more stable under normoxia than WT RAP2.12, and also that the dynamics of its stabilisation under hypoxia are different. It would be useful if the authors could include a quantification of the stability of these constructs. E.g. in Fig3C, the signal at 21% oxygen (i.e., untreated) seems to be quite variable at 1, 2 and 4h timepoints (but should be consistent?), and the alternate blot in Fig S8B does not show the same pattern. Could the authors clarify/expand on these discrepancies further, and also provide extra repeats?

The authors should do RNA-seq, rather than qPCR of only 2 hypoxia marker genes (SAD6 and PCO1) to get a broader overview of the effects of the HIFOOD-RAP2.12 in the *erf1ll* pentuple background. This would be useful to show how global transcription is affected.

Could the authors provide some further rationale as to why the expression patterns of HRGs were modelled? What was used as the basis of the modelling – i.e., what experimental data was it based on? I found this part of the manuscript less easy to follow. Also, again in this section of the manuscript, only two genes are used as a read out of the system – why only two? The hypoxia survival data in this section are very nice, however, and clearly show that the synthetic O₂ transduction system is sufficient to decrease the lethal effects of hypoxia.

The engineering of PIF4 as a hypoxia regulated protein for the induction of an elongation response under submergence is a very interesting and exciting idea. The data here are also very nice and show that the function of the PIF4OOD construct could be dampened by PHD3 and enhanced by submergence, to promote petiole elongation. However, here the authors need to show the levels of PIF4 transcript and protein, and show that they correlate with the phenotypes: i.e., in the presence of PHD the protein levels should be relatively depleted, and contrastingly the protein levels should accumulate under submergence (all without changes in transcript level). It appears that the PIF4 constructs aren't tagged, but this should not be an issue as several PIF4 antibodies (including commercially available ones) exist.

I really liked the discussion section, particularly because it honestly referred to the challenges faced during the development and optimisation of the tools presented.

Reviewer #4

(Remarks to the Author)

This study engineered a synthetic oxygen-sensing system in *Arabidopsis thaliana* to mimic animal-like hypoxia responses. The system partially restored hypoxia responsiveness in oxygen-insensitive mutants and contributed to the regulation of plant growth under flooding conditions. The work demonstrates the power of synthetic biology to reprogram signaling pathways, offering insights into the evolution of oxygen sensing and potential applications for improving crop tolerance to stress. While the study is compelling, several issues should be addressed to strengthen the manuscript.

Major:

- 1) The study uses FLuc enzymatic activity as a proxy for hypoxia response. Have the authors measured FLuc mRNA levels under these conditions to confirm that changes in luciferase activity are not due to transcriptional regulation?
- 2) For the assays presented in Figures 2B and 2C, have the corresponding transcript levels (e.g., via qRT-PCR) been quantified to support the protein activity data?
- 3) From a systems biology perspective, the mathematical modeling (Fig. 4A and Online Methods) lacks a strong connection to the experimental results. While the technical execution is sound, the analysis does not appear to motivate new experiments or offer explanatory insights into observed phenomena. As it stands, the modeling seems dispensable unless its biological relevance is more clearly established.
- 4) The model incorporates parameters both from the literature and from data fitting. How do the authors ensure that this hybrid approach yields a model that generalizes across different experimental conditions?
- 5) In Figure 5, does "WT" refer to PIF4 plants that underwent dephosphorylation but were not treated with HIFodd (as shown in Fig. 5A)? Under these conditions, do PIF4 and HIFodd still interact with the CULLIN3–BOP2 complex? How can it be confirmed that oxygen is the only variable influencing the observed effects?

Minor:

- 1) The loading control in Figure 1F is unclear. Please provide the original image or replace it with a clearer version.
- 2) In the Results section titled "Exploiting PHD-dependent proteolysis to engineer adaptive responses to flooding in plants," the cited supplementary figure numbers are incorrect. S11 should be updated to S12, and S12 to S13.
- 3) The expression levels in Figure 4A are shown without units, while both the modeling analysis and Figure S10 include them. Please add appropriate units for consistency and clarity.

- 4) The bars in Supplementary Figure S10 are not explained. Please include a legend or description.
- 5) The control phenotype for Line 1 in Figure 5 does not align well with the corresponding bar chart. Consider replacing it with a more representative image.
- 6) The manuscript contains several typographical and formatting errors, including an incorrect email address. A thorough proofreading is recommended prior to resubmission.

Reviewer #5

(Remarks to the Author)

Version 1:

Reviewer comments:

Reviewer #1

(Remarks to the Author)

I am satisfied that most of my comments have been addressed.

Please can the authors check ALL the references carefully as some of them appear to be incorrect (e.g. 61, 86, 88, maybe 70-72?)

Please could the plus and minus symbols be made larger in Fig. S14F?

"petiole length" is spelled incorrectly in some of the supplementary figures

(Remarks on code availability)

Reviewer #2

(Remarks to the Author)

Thanks for addressing all of my critiques, in particular including the complete Western with loading control (Fig 1F) and clarifying and fixing the unit conversions for the model. The responses have addressed all concerns.

(Remarks on code availability)

Reviewer #3

(Remarks to the Author)

I confirm that I am satisfied with the revised manuscript submitted to Nature Communications, and that all of my queries and concerns have been appropriately addressed in the revision.

(Remarks on code availability)

Reviewer #4

(Remarks to the Author)

I do not have any further comment.

(Remarks on code availability)

I do not have any further comment.

Reviewer #5

(Remarks to the Author)

(Remarks on code availability)

Yes. The code is publicly available and well organized, and it includes a comprehensive README with step-by-step instructions for environment setup, dependency installation, and running the main experiments. The repository provides example commands/scripts, configuration files, and the necessary data-processing pipeline. I was able to install and run the

code without major issues. Using the provided scripts, I reproduced the main results and trends reported in the manuscript (including key figures/tables and evaluation metrics). Minor deviations, if any, are within expected numerical variability (e.g., due to random seeds, hardware/software versions). Overall, the code appears to be a usable and valuable resource for the community and should support reproducibility.

We are grateful to the five reviewers for their thorough scrutiny of our manuscript, their supportive comments, and their constructive criticism, which have helped us to improve the study.

Below, we provide a point-by-point response to their comments. Revisions made in response to the reviewers' concerns are indicated by line numbers at each point and are highlighted in blue in the revised manuscript.

Additional comment:

While revising the manuscript, we realized two additional inaccuracies that are not mentioned in the reviewers' comments. We apologize for these mistakes, which have now been corrected in the current version of the manuscript:

1. We identified a unit error in one parameter (the activation rate for HRG mRNA) and corrected it from μM to nM in Supplementary Table S6 and S7.
2. The construct described in Fig. 5 is actually HIF_{ODD}-PIF4 and not PIF4-HIF_{ODD}. We corrected this in the figure and in the text.

Replies to Reviewer 1's comments

1. Surprisingly, RAP2.12 bearing the HIF1 degron was much more stable in normoxia than hypoxia after 4 h (Fig. 3C and Fig. 8B) suggesting that protein abundance becomes uncoupled from O₂ availability. This remains to be explained; the authors speculate that other post-translational mechanisms operate and there is evidence for these in the literature, but it is not clear why they would differentially affect the synthetic RAP2.12 to destabilize it in hypoxia.

Re: The reviewer is correct. Although we do not currently have a definitive explanation for this repeated observation, we speculate that it arises from crosstalk between proteolytic regulatory pathways that control RAP2.12 abundance in response to light and oxygen. Notably, the concomitant abolition of N-degron-dependent regulation (and potentially the addition of the HIF domain) appears to interfere with a yet uncharacterised light-dependent pathway, resulting in the stabilisation of RAP2.12. We are interested in investigating this light-dependent regulation in future studies, with a more specific focus on the endogenous regulation of ERFVILs.

We have conducted experiments to investigate the presence of a cryptic, non-oxygen-dependent internal N-degron in RAP2.12 by developing a new strategy based on the ZZ domain of the human p62 protein, inspired by the pioneering work of Seo et al. (2023, [10.1007/s00018-021-03805-x](https://doi.org/10.1007/s00018-021-03805-x)). In short, we observed that removing the first 13 amino acid of RAP2.12 does not abolish recognition by PRT6. We are planning to submit this as a separate manuscript as this will diverge our story too far from the scope of the study reported here.

2. This is an exciting result but is lacking information about any potential negative impacts on the plants (e.g. the pRON3:PIF4-HIF_{ODD} plants appear to have fewer leaves than the wild type) which would be relevant to translating this system to crops.

Re: We thank the reviewer for raising this point. It is true that the pRON3:PIF4-HIF_{ODD} plants shows an increased plastochron and accelerated flowering. We included in the manuscript a sentence to describe and comment on this phenotype (**lines 409-412**), which is expected when increasing PIF4 levels, in agreement with other reports (e.g. Jenkitkonchai et al. 2021, [10.1002/pld3.339](https://doi.org/10.1002/pld3.339)).

Whilst the work represents an impressive feat of synthetic biology, I am less convinced that the authors have fully answered the evolutionary question that they posed.

Re: We thank the reviewer for their appreciation of our work. We also agree with them that the evolutionary question of ‘*why animals and plants rely on different dioxygenases to control the stability of transcription factors regulating gene expression in response to hypoxia*’ remains far from fully resolved. However, we hope we successfully convinced the reviewer that our approach does demonstrate that the two systems are, in fact, interchangeable across kingdoms. This finding paves the way for future efforts, for example through accelerated evolution, to match the gene expression dynamics governed by endogenous systems. We have made this point explicit in the Discussion (**lines 524-529**).

Presumably animal cells use the two systems in different contexts. The manuscript could be improved by considering this and also the different ways in which plants and animals encounter hypoxia.

Re: We thank the reviewer for the suggestion. In metazoans, hypoxia responses involve the Cys branch of the N-degron pathway, which however regulates non-nuclear processes and it has been proposed to enable rapid responses to decreased O₂ levels. In contrast, there is no direct evidence that PHD enzymes regulate protein stability in plants, although indirect roles in plant hypoxia responses have been suggested. Differences in hypoxia-regulated transcription between plants and animals likely reflect how each experiences oxygen limitation: animals face damaging acute hypoxia despite circulatory oxygen delivery, whereas plants lack active oxygen transport, tolerate short-term hypoxia better, and often experience hyperoxia due to photosynthesis. We included a discussion of these two points at **lines 488-508**.

Fig. 4 was the survival and root growth experiment performed more than once?

Yes, this was repeated twice with similar results. This additional set of data is now shown in the new **Supplementary Figure 12**.

Was the O₂ concentration response of the PCO and SAD6 expression modelled?

Re: We thank the reviewer for suggesting this addition. We modelled *HRG* expression as a function of different hypoxia levels. This is now added as **Fig. 4 (B)** and discussed at **lines 326-332**. We would like to flag, however, that this should not be interpreted as a model of the response dynamics of *HRGs* to decreasing oxygen through time, but as a determination of *HRG* expression at different oxygen steady states.

“hypoxia-inducible version of PIF4” could imply that a hypoxia-responsive promoter was used. Perhaps rephrase to state “oxygen sensitive version of PIF4” or “hypoxia-stabilised version of PIF4”?

Re: We agree that inducible is more frequently interpreted as referred to mRNA regulation. We changed it in the manuscript as suggested (**line 396**).

Fig. 5: were PIF4 protein levels quantified?

Yes, we now included the quantification of HIF-PIF4 protein and mRNA levels in **Fig. 5 (B-C)** and **Supplementary Fig. 14**. This is now discussed at **lines 408-409**.

Replies to Reviewer 2's comments

1. For the Western blots reported, the loading controls on gels (Fig 1F, SI except for the mammalian cell line experiment) appear spotty in some cases; it would benefit the paper to redo these experiments. The conclusions of expression at different environmental conditions and time points rest on these experiments. Additionally, the unmodified gels must be deposited in Zenodo or other DOI-linked resource for fair evaluation.

Re: We managed to improve detection of HIF_{ODD}-GAL4 (**Fig. 1F**) using a FLAG antibody instead of the GAL4 antibody. Detection of RAP2.12 proteins from plant samples is very tricky in our hands, and according to discussions at the recent International Society of Plant Low Oxygen Research this is true across labs. Indeed, these are the blots where we can see bands that we can confidently consider corresponding to RAP2.12 protein.

Unmodified gels have been deposited in Zenodo (10.5281/zenodo.18074079). Access to this dataset is currently restricted by an embargo until 2 April 2026. However, the same document is uploaded for the reviewers and the Nat Comm editorial team (Uncropped Western Blots).

2. The definition of relative Fluc activity (Figure 2B) is not discussed in the methods or the main results section (unless I missed it). I'm confused because the line #1 shows a baseline of relative Fluc at 2, and I would have expected 1.0. Please clarify.

Re: We thank the reviewer for giving us the opportunity to clarify how relative Fluc was calculated. We used the ratio between the HIF_{ODD}-linked luciferase and the HIF_{ODD}-independent luciferase. Depending on the luminescence properties of the enzyme pair, in our experimental setup the baselines can vary below 1 and above 2 (**Fig.1** and **S5**). We included this sentence in M&M (lines **678-679**).

3. For the mathematical modeling section, I'm a bit surprised that the model showed high micromolar steady state concentrations of the transcription factor proteins. I couldn't tell from Supp Tables 3-5 as to whether the equations are dimensionally homogeneous; if not a unit conversion mistake, the authors should comment on the validity and limitations of their model considering the predicted concentrations.

Re: We thank the reviewer for this observation and the opportunity to clarify. The RAP2.12 protein concentrations in Supplementary Table 5 are the steady-state values under normoxia (20% O₂), which were then used as initial conditions for the hypoxia simulations (1% O₂). These values are in the nM to sub- μ M range, not in the high micromolar range. For example:

- RAP2.12 = 0.0009 μ M = 0.9 nM
- HIF_{ODD}- Δ RAP2.12 = 0.0058 μ M = 5.8 nM
- HRG mRNA (corrected) = $7.2 \times 10^{-5} - 4.4 \times 10^{-4}$ μ M (0.072–0.44 nM)

During revision we identified and corrected a unit error in the mRNA synthesis rate (k_4), which should be expressed in nM \cdot s⁻¹ rather than μ M \cdot s⁻¹. This correction reduces the absolute steady-state values of HRG mRNA by 1000-fold (as listed above) but does not alter the *patterns* of hypoxic response or our comparative conclusions, since the same synthesis parameters were applied consistently in both the PCO4/RAP2.12 and PHD/HIF_{ODD}- Δ RAP2.12 models.

Regarding dimensional homogeneity:

- PCO4 and PHD, and Michaelis–Menten constants (Kms) are expressed in μM (except Kms for O_2 in %).
- Time is in seconds (rate constants k in s^{-1} , except k_1 and k_4 in $\text{nM}\cdot\text{s}^{-1}$).
- Fluxes (v_1 – v_9) therefore have units of $\mu\text{M}\cdot\text{s}^{-1}$.
- For example, in the key equation describing RAP2.12 degradation by PCO4:
$$v_3 = k_3 \times [\text{PCO4}] \times [\text{O}_2]/(\text{Km}_1 + [\text{O}_2]) \times [\text{RAP2.12}]/(\text{Km}_2 + [\text{RAP2.12}])$$
Here, $[\text{O}_2]/(\text{Km}_1 + [\text{O}_2])$ and $[\text{RAP2.12}]/(\text{Km}_2 + [\text{RAP2.12}])$ are dimensionless ratios. Thus, the only terms carrying units are k_3 (s^{-1}) and $[\text{PCO4}]$ (μM), which together yield v_3 in $\mu\text{M}\cdot\text{s}^{-1}$. This demonstrates that the equations are dimensionally homogeneous.

For clarity, although the framework is parameterized in μM , we now report the simulated steady-state concentrations of HRG mRNA in nM in Supplementary Table 5, since their absolute values are sub-nM after correction of the synthesis rate.

In terms of validity and limitations, our model is a simplified kinetic representation. The predicted steady-state concentrations are within a biologically reasonable range, but the absolute values should be interpreted with caution because:

- Protein and mRNA synthesis rates (k_1 , k_4) were estimated from the literature and are not cell-specific. However, because the same synthesis parameters were applied in both models, this does not bias the comparison. The simulated hypoxic response is determined primarily by enzyme–substrate kinetics rather than synthesis assumptions.
- The model does not explicitly include additional regulatory factors (e.g., protein–protein interactions, compartmentalization, cell division) that could further modulate concentrations *in vivo*.

Therefore, while the model is dimensionally consistent and produces physiologically plausible concentration ranges, its main purpose is to capture the *relative* dynamics and oxygen-dependent regulation of the two systems rather than to predict absolute concentrations with high precision. We have revised the manuscript accordingly to clarify both the corrected units and the limitations of our modelling framework.

4. Small – the UBQ fusion protein is a clever approach. Most use variations of a 2A peptide. Is there a reference for this UBQ fusion strategy?

Re: Thank you for the suggestion. We included Bachmair et al. 1986 ([10.1126/science.301893](https://doi.org/10.1126/science.301893)) as the reference for this strategy.

REVIEWER 3.

It would be useful if the authors could include a quantification of the stability of these constructs. E.g. in Fig3C, the signal at 21% oxygen (i.e., untreated) seems to be quite variable at 1, 2 and 4h timepoints (but should be consistent?), and the alternate blot in Fig S8B does not show the same pattern. Could the authors clarify/expand on these discrepancies further, and also provide extra repeats?

Re: The reviewer is exactly right about the variable stability of *HIF_{OOD}-RAP2.12*. We were too puzzled by the stabilisation observed after 4h in the dark. We repeated the experiment a third time (up to 6h hypoxia or normoxia in the dark) with a similar outcome (**Supplementary Fig. 8C**). We

suspect the contribution of prolonged darkness to a (transient) stabilisation linked to a feedback mechanism. An effect of light on the (de)stabilisation of ERFVIs has been reported before (Abbas et al., 2015, [10.1016/j.cub.2015.03.060](https://doi.org/10.1016/j.cub.2015.03.060)). However, we also feel that investigation of this additional regulation should be more appropriately addressed with a dedicated set of experiments focusing on the wild-type ERFVIs. We included one additional experiment to confirm that, at least in the early phases of hypoxia treatments, the *HIF_{ODD}-RAP2.12* protein is under control of the proteasome in a PHD3-dependent manner (**Supplementary Fig. 8D**).

In the light of several review articles on the western blot method (e.g. [10.1002/elps.200800720](https://doi.org/10.1002/elps.200800720) and [10.1126/scisignal.2005966](https://doi.org/10.1126/scisignal.2005966)), we respectfully argue against the use of this technique as a quantitative method when using independently acquired repetitions (as in our case).

The authors should do RNA-seq, rather than qPCR of only 2 hypoxia marker genes (SAD6 and PCO1) to get a broader overview of the effects of the HIF_{ODD}-RAP2.12 in the erfvi pentuple background. This would be useful to show how global transcription is affected.

Re: Following the reviewer's advice, we ran an RNA-seq of HIF_{ODD}-RAP2.12 plants treated with hypoxia (1% and 0.1% O₂). Overall, approximately 400 genes were significantly up-regulated under both conditions, including 85% of the genes identified as core-response to hypoxia (Mustroph et al. 2009, [10.1073/pnas.0906131106](https://doi.org/10.1073/pnas.0906131106)). We also analysed this data set comparing it with existing data from the *erfvi* and wild type exposed to the same hypoxic conditions (1%, 2h) and published before in Dalle Carbonare et al. 2025 ([10.1016/j.molp.2025.05.015](https://doi.org/10.1016/j.molp.2025.05.015)). These data are presented in **Supplementary Tables S1-3**. In the manuscript, we focus on the core 49 HRGs (**Fig. 4D**) and discussed that a majority of these genes are significantly higher than in the *erfvi* background but also significantly induced by hypoxia (lines **342-350**).

Could the authors provide some further rationale as to why the expression patterns of HRGs were modelled? What was used as the basis of the modelling – i.e., what experimental data was it based on? I found this part of the manuscript less easy to follow. Also, again in this section of the manuscript, only two genes are used as a read out of the system – why only two?

Re: Our modelling approach aimed to predict the effect of HIF_{ODD}-RAP2.12 expression in plant cells when these are exposed to hypoxia (before running the actual experiment). We focussed on two transcripts, among the 49 core hypoxia response genes, that in the literature are described as primarily regulated by ERFVIs and, for example, not by other TFs regulated by hypoxia (Eysholdt-Derzso et al. 2023, [10.1073/pnas.2221308120](https://doi.org/10.1073/pnas.2221308120) and Dalle Carbonare et al. 2025 [10.1016/j.molp.2025.05.015](https://doi.org/10.1016/j.molp.2025.05.015)). The model is not created by fitting these or other existing transcript datasets, but rather on the biochemical parameters available for the various components that constitute the ERFVI-HRPE module.

The authors need to show the levels of PIF4 transcript and protein.

Re: We included a western blot and realtime qPCR quantification of PIF4-HIF protein and mRNA, respectively (**Figure 5B-C** and **Supplementary Figure S13-S14**).

Replies to reviewer 4.

1) *Have the authors measured FLuc mRNA levels under these conditions to confirm that changes in luciferase activity are not due to transcriptional regulation?*

2) For the assays presented in Figures 2B and 2C, have the corresponding transcript levels (e.g., via qRT-PCR) been quantified to support the protein activity data?

Re: We thank the reviewer for giving us the opportunity to clarify our experimental setup. In the analyses described with **Figure 1D-E**, the regulation of FLuc by HIF-Gal4 is indeed transcriptional. In transient transactivation assay with Arabidopsis protoplasts, FLuc activity is commonly used as the output of a transcriptional reporter. In the experiments shown in **Figure 2B-D**, instead, the same promoter controls the expression of the two luciferases. The FLuc associated with HIF_{ODD} and the HIF-independent RLuc, which is hypoxia independent, are synthesized as a single polypeptide and then separated by DUBs during translation. In this case, the regulation can only occur at the post-translational level.

3) From a systems biology perspective, the mathematical modeling (Fig. 4A and Online Methods) lacks a strong connection to the experimental results. While the technical execution is sound, the analysis does not appear to motivate new experiments or offer explanatory insights into observed phenomena. As it stands, the modeling seems dispensable unless its biological relevance is more clearly established.

Re: We thank the reviewer for this important comment. Our modelling in **Fig. 4A-B** was designed as an *in silico* framework to predict and compare the dynamics of oxygen sensing through the plant PCO4/RAP2.12 system and the animal PHD/HIF_{ODD}- Δ RAP2.12 system, based on published kinetic data. The model predicted that the PHD/HIF_{ODD}- Δ RAP2.12 pathway would produce a stronger hypoxic response than the endogenous PCO4/RAP2.12 pathway in plants.

This prediction offered explanatory value and shaped the future research. It raised the hypothesis that synthetic introduction of an animal-type oxygen-sensing module into plants could lead to stronger hypoxic responses and distinct phenotypes, particularly under hypoxia or submergence. The model also predicted that the relative strengths and timings of the two systems would be governed primarily by enzyme–substrate kinetics, which both provided a rationale for our experimental approach, including testing the synthetic or hybrid oxygen-sensing circuits in planta to examine how altering the oxygen sensor can reshape stress adaptation.

Thus, the modelling is not intended as a standalone result but as a comparative tool that guided and complements our experiments and, in the future, can generate testable predictions about oxygen sensing and potential synthetic applications in plant and animal cells.

4) The model incorporates parameters both from the literature and from data fitting. How do the authors ensure that this hybrid approach yields a model that generalizes across different experimental conditions?

Re: The kinetic assays for both PCO4 and PHD2 were performed by the same laboratory (Emily Flashman's group). This provides consistency in assay methodology and quality control across the two systems, thereby reducing variability that might arise if data had been collected by different groups using unrelated protocols.

We acknowledge, however, that there remain inherent experimental differences between the plant PCO4 and mammalian PHD2 kinetic assays (e.g. temperature of 25 °C versus 37 °C, and use of plant versus mammalian reaction buffer). Because of this, the absolute catalytic values are not directly comparable across species, and our model should not be interpreted as a fully quantitative generalization.

Our intention is to place both enzyme systems into an *idealized standardized environment in silico*, to allow a fair comparison of their catalytic properties. Even under these conditions, the PHD system shows lower catalytic efficiency than the PCO system, consistent with experimental observations *in vivo*. This makes the comparison biologically meaningful, and it raises the intriguing question of how a less efficient oxygen-sensing enzyme such as PHD would function in a plant background and affect hypoxia- or submergence-related phenotypes.

We have revised the Method to emphasize both the limitations and its value as a hypothesis-generating comparison.

5) In Figure 5, does "WT" refer to PIF4 plants that underwent dephosphorylation but were not treated with HIF_{odd} (as shown in Fig. 5A)? Under these conditions, do PIF4 and HIF_{odd} still interact with the CULLIN3–BOP2 complex? How can it be confirmed that oxygen is the only variable influencing the observed effects?

Re: Thank you for this comment, we corrected the figure by indicating the wild type as Col-0, to avoid the confusing effect that HIF_{odd}-PIF4 is a non-phosphorylatable version of the transcription factor. With the western blot suggested by reviewer 2 and 3, we provide evidence that hypoxia can stabilise the chimeric HIF_{odd}-PIF4 (**Fig. 5B** in the new version). However, the reviewer is right that we cannot exclude additional effects of submergence impacting on HIF_{odd}-PIF4. Therefore, we included a sentence to describe this experimental limitation in the manuscript (**lines 421-422**).

Minor

1) The loading control in Figure 1F is unclear. Please provide the original image or replace it with a clearer version.

Re: The figure has been replaced with a new blot. As we explained in our reply to Reviewer 1, the anti-Gal4 antibody did not work well in our hands, therefore we used an anti-FLAG antibody.

2) In the Results section titled "Exploiting PHD-dependent proteolysis to engineer adaptive responses to flooding in plants," the cited supplementary figure numbers are incorrect. S11 should be updated to S12, and S12 to S13.

Re: We thank the reviewer for spotting this mistake. We corrected the numbering of the Supplementary Figures.

3) The expression levels in Figure 4A are shown without units, while both the modeling analysis and Figure S10 include them. Please add appropriate units for consistency and clarity.

Re: We thank the reviewer for this helpful suggestion. The values in Figure 4A are expressed in μM , consistent with the parameterization of the kinetic model. However, these should be interpreted as simulated concentrations of a generic HRG mRNA, not as exact *in vivo* levels in plants. The absolute values are limited by assumptions about compartment volume, omission of additional regulatory factors, and the fact that synthesis rates (k_1 , k_4) were chosen as reasonable literature-based parameters rather than cell-specific measurements. Importantly, the same synthesis parameters were applied in both the PCO4/RAP2.12 and PHD/HIF_{odd}- Δ RAP2.12 models, so the purpose of Fig. 4A is to enable a fair comparison of response dynamics, not to predict the precise μM concentration of each species.

To address the reviewer's concern, we have revised Figure 4A to label the y-axis as "Simulated HRG expression (nM)" and updated the legend to clarify that these values are model predictions under simplified assumptions, included for relative comparison only. This ensures consistency with **Figure S10** while preventing misinterpretation that the model directly measures cellular concentrations.

4) The bars in Supplementary Figure S10 are not explained. Please include a legend or description.

Re: We thank the reviewer for suggesting this clarification. The error bars represent standard errors (SE), as reported in the original paper. This is now described in the figure legend.

5) The control phenotype for Line 1 in Figure 5 does not align well with the corresponding bar chart. Consider replacing it with a more representative image.

Re: The image has been replaced as suggested.

6) The manuscript contains several typographical and formatting errors, including an incorrect email address. A thorough proofreading is recommended prior to resubmission.

Re: Thank you for suggesting this. We revised the manuscript in its current form to the best of our capacity. We will rely on the editorial teams at Nature comms to refine the text according to their standards.

Reply Reviewer #1

Please can the authors check ALL the references carefully as some of them appear to be incorrect (e.g. 61, 86, 88, maybe 70-72?)

We thank the reviewer for this suggestion which has prompted us to check our references and find three additional incorrect citations, in addition to those they listed. All of the references have been corrected now.

Please could the plus and minus symbols be made larger in Fig. S14F?

We increased the size of the plus and minus symbols.

"petiole length" is spelled incorrectly in some of the supplementary figures.

This has been corrected.

Reviewer #5 (Remarks on code availability):

We are especially thankful to this reviewer for testing our code.